# Decomposition Ascribed Synergistic Learning for Unified Image Restoration

## Abstract

Learning to restore multiple image degradations within a single model is quite beneficial for real-world applications. Nevertheless, existing works typically concentrate on regarding each degradation independently, while their relationship has been less comprehended to ensure the synergistic learning. To this end, we revisit the diverse degradations through the lens of singular value decomposition, with the observation that the decomposed singular vectors and singular values naturally undertake the different types of degradation information, dividing various restoration tasks into two groups, *i.e.*, singular vector dominated and singular value dominated. The above analysis renders a more unified perspective to ascribe diverse degradation connections, compared to previous task-level independent learning. The dedicated optimization of degraded singular vectors and singular values inherently utilizes the potential partnership among diverse restoration tasks, attributing to the Decomposition Ascribed Synergistic Learning (DASL). Specifically, DASL comprises two effective operators, namely, Singular VEctor Operator (SVEO) and Singular VAlue Operator (SVAO), to favor the decomposed optimization, which can be lightly integrated into existing image restoration backbone. Moreover, the congruous decomposition loss has been devised for auxiliary. Extensive experiments on five image restoration tasks demonstrate the effectiveness of our method.

## 1 Introduction

Image restoration aims to recover the latent clean images from their degraded observations, and has been widely applied to a series of real-world scenarios, such as photo processing, autopilot, and surveillance. Compared to single-degradation removal Zhou et al. (2021a); Xiao et al. (2022); Qin et al. (2020); Song et al. (2023); Lehtinen et al. (2018); Lee et al. (2022); Pan et al. (2020); Nah et al. (2021); Li et al. (2023); Zhang et al. (2022), the recent flourished multi-degradation learning methods have gathered considerable attention, due to their convenient deployment. However, every rose has its thorn. How to ensure the synergy among diverse restoration tasks demands a dedicated investigation, and it is imperative to comprehend the property of the involved degradations judiciously and include their implicit relationship into consideration.

Generally, existing multi-degradation learning methods concentrated on regarding each degradation independently. For instance, Chen et al. (2021); Li et al. (2020); Valanarasu et al. (2022) propose to deal with different restoration tasks through separate subnetworks or distinct transformer queries. Li et al. (2022); Chen et al. (2022b) propose to distinguish diverse degradation representations via contrastive learning. Remarkably, there are also few attempts devoted to duality degradation removal with synergistic learning. Zhang et al. proposes to leverage the blurry and noisy pairs for joint restoration as their inherent complementarity during digital imaging. Zhou et al. (2022b) proposes a unified network with low-light enhancement encoder and deblurring decoder to address hybrid distortion. Wang et al. (2022a) proposes to quantify the relationship between arbitrary two restoration tasks, and improve the performance of the anchor task with the aid of another task. However, few efforts have been made toward the synergistic learning among more restoration tasks, and there is desperately lacking of a general perspective to comprehend diverse degradations for combing their implicit connections, which set up the stage for this paper.

To solve the above problem, we propose to revisit diverse degradations through the lens of singular value decomposition, and conduct experiments on five common image restoration tasks, including

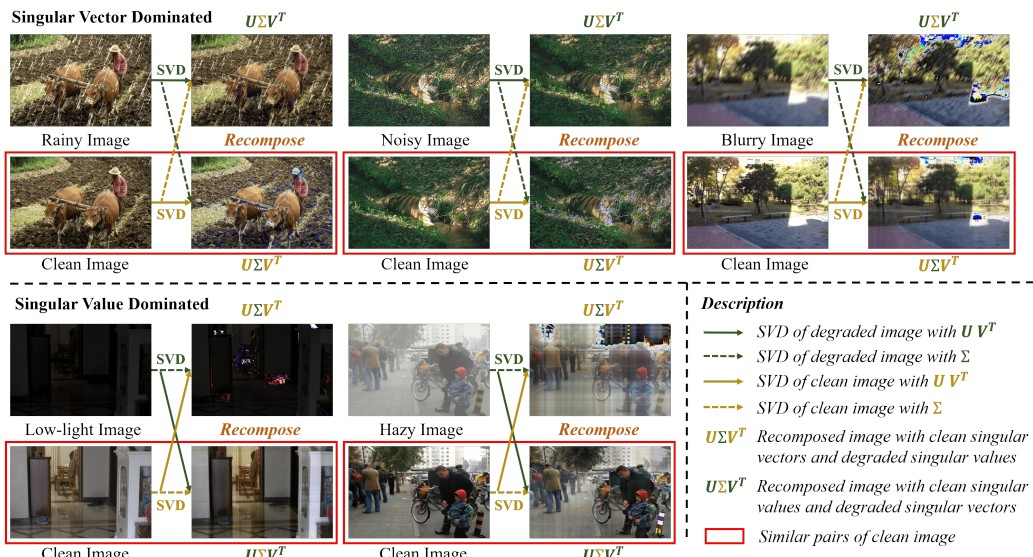

Figure 1: An illustration of the decomposition ascribed analysis on various image restoration tasks through the lens of the singular value decomposition. The decomposed singular vectors and singular values undertake the different types of degradation information as we recompose the degraded image with portions of the clean counterpart, ascribing diverse restoration tasks into two groups, *i.e.*, singular vector dominated *rain, noise, blur*, and singular value dominated *low-light, haze*. Dedicated to the decomposed optimization of the degraded singular vectors and singular values rendering a more unified perspective for synergistic learning, compared to previous task-level independent learning.

image deraining, dehazing, denoising, deblurring, and low-light enhancement. As shown in Fig. 1, it can be observed that the decomposed singular vectors and singular values naturally undertake the different types of degradation information, in that the corruptions fade away when we recompose the degraded image with portions of the clean counterpart. Thus, various restoration tasks can be ascribed into two groups, *i.e.*, singular vector dominated degradations and singular value dominated deagradations. The statistic results in Fig. 2 further validate this phenomenon, where the quantified comparison of the recomposed image quality and singular distribution discrepancy have been presented. Therefore, the potential partnership emerged among diverse restoration tasks could be inherently utilized through the decomposed optimization of singular vectors and singular values, considering their ascribed common properties. Note that more other degradation analyses and theoretical generalization verification are provided in Appendix G.

In this way, we decently convert the previous task-level independent learning into more unified singular vectors and singular values learning, and form our method, Decomposition Ascribed Synergistic Learning (DASL). Basically, one straightforward way to implement our idea is to directly perform the decomposition on latent high-dimensional tensors, and conduct the optimization for decomposed singular vectors and singular values, respectively. However, the huge computational overhead is non-negligible. To this end, two effective operators have been developed to favor the decomposed optimization, namely, Singular VEctor Operator (SVEO) and Singular VAlue Operator (SVAO). Specifically, SVEO takes advantage of the fact that the orthogonal matrices multiplication makes no effect on singular values and only impacts singular vectors, which can be realized through simple regularized convolution layer. SVAO resorts to the signal formation homogeneity between Singular Value Decomposition and the Inverse Discrete Fourier Transform, which can both be regarded as a weighted sum on a set of basis. While the decomposed singular values and the transformed fourier coefficients inherently undertake the same role for linear combination. And the respective base components share similar principle, *i.e.*, from outline to details. Therefore, with approximate derivation, the unattainable singular values optimization can be translated to accessible spectrum maps. We show that the fast fourier transform is substantially faster than the singular value decomposition. Furthermore, the congruous singular decomposition loss has been devised for auxiliary. The proposed DASL can be lightly integrated into existing image restoration backbone for decomposed optimization.

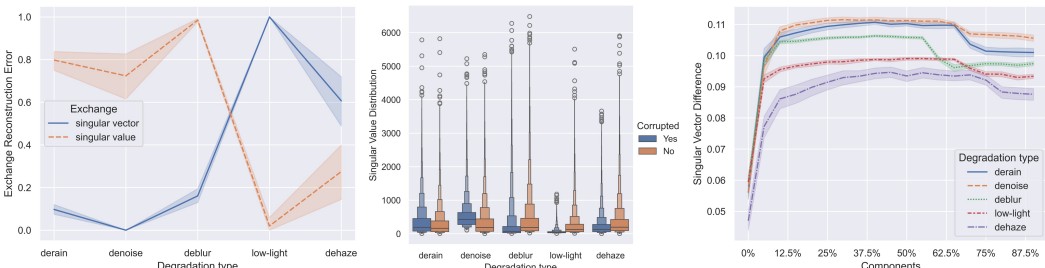

Figure 2: The statistic validation that the decomposed singular vectors and singular values undertake different types of degradation information. (a) The reconstruction error between the recomposed image and paired clean image on five common image restoration tasks. Low error denotes the degradation primarily distributed in the replaced portion of the image. (b) The boxplot comparison of singular value distribution between the degraded image and corresponding clean image, where the singular value dominated *low-light* and *haze* exhibit extraordinary difference. (c) The singular vector difference on separate orders of the component between the degraded image and clean image, where the singular vector dominated *rain*, *noise*, and *blur* present more disparity. The results are obtained under calculation on ∼1k images for each restoration task.

The contributions of this work are summarized below:

- We take a step forward to revisit the diverse degradations through the lens of singular value decomposition, and observe that the decomposed singular vectors and singular values naturally undertake the different types of degradation information, ascribing various restoration tasks into two groups, *i.e.*, singular vector dominated and singular value dominated.

- We propose the Decomposition Ascribed Synergistic Learning (DASL) to dedicate the decomposed optimization of degraded singular vectors and singular values respectively, which inherently utilizes the potential partnership among diverse restoration tasks.

- Two effective operators have been developed to favor the decomposed optimization, along with a congruous decomposition loss, which can be lightly integrated into existing image restoration backbone. Extensive experiments on five image restoration tasks demonstrate the effectiveness of our method.

## 2 RELATED WORK

**Image Restoration.** Image restoration aims to recover the latent clean images from degraded observations, which has been a long-term problem. Traditional image restoration methods typically concentrated on incorporating various natural image priors along with hand-crafted features for specific degradation removal Babacan et al. (2008); He et al. (2010); Kundur & Hatzinakos (1996). Recently, learning-based methods have made compelling progress on various image restoration tasks, including image denoising Lehtinen et al. (2018); Lee et al. (2022), image deraining Zhou et al. (2021a); Xiao et al. (2022), image deblurring Pan et al. (2020); Nah et al. (2021), image dehazing Zheng et al. (2021); Song et al. (2023), and low-light image enhancement Li et al. (2023); Guo et al. (2020), *etc*. Moreover, numerous general image restoration methods have also been proposed. Zamir et al. (2021; 2022a); Fu et al. (2021) propose the balance between contextual information and spatial details. Mou et al. (2022) formulates the image restoration via proximal mapping for iterative optimization. Zhou et al. (2022a; 2023) proposes to exploit the frequency characteristics to handle diverse degradations. Additionally, various transformer-based methods Zamir et al. (2022b); Liu et al. (2022); Liang et al. (2021); Wang et al. (2022c) have also been investigated, due to their impressive performance in modeling global dependencies and superior adaptability to input contents.

Recently, recovering multiple image degradations within a single model has been coming to the fore, as they are more in line with real-world applications. Zhang et al. proposes to leverage the short-exposure noisy image and the long-exposure blurry image for joint restoration as their inherent complementarity during digital imaging. Zhou et al. (2022b) proposes a unified network to address low-light image enhancement and image deblurring. Furthermore, numerous all-in-one fashion methods Chen et al. (2021); Li et al. (2020; 2022); Valanarasu et al. (2022); Chen et al. (2022b) have been proposed to deal with multiple degradations. Zhang et al. (2023) proposes to correlate various

degradations through underlying degradation ingredients. While Park et al. (2023) advocates to separate the diverse degradations propcessing with specific attributed discriminative filters. Besides, most of existing methods concentrated on the network architecture design and few attempts have been made toward exploring the synergy among diverse image restoration tasks.

**Tensor Decomposition.** Tensor decomposition has been widely applied to a series of fields, such as model compression Jie & Deng (2022); Obukhov et al. (2020), neural rendering Obukhov et al. (2022), multi-task learning Kanakis et al. (2020), and reinforcement learning Sozykin et al.. In terms of image restoration, a large number of decomposition-based methods have been proposed for hyperspectral and multispectral image restoration Peng et al. (2022); Wang et al. (2020; 2017), in that establishing the spatial-spectral correlation with low-rank approximation.

Alternatively, a surge of filter decomposition methods toward networks have also been developed. Zhang et al. (2015); Li et al. (2019); Jaderberg et al. (2014) propose to approximate the original filters with efficient representations to reduce the network parameters and inference time. Kanakis et al. (2020) proposes to reparameterize the convolution operators into a non-trainable shared part and several task-specific parts for multi-task learning. Sun et al. (2022) proposes to decompose the backbone network and only finetune the singular values to preserve the pre-trained semantic clues for few-shot segmentation.

## 3 METHOD

In this section, we start with introducing the overall framework of Decomposition Ascribed Synergistic Learning in Section 3.1, and then elaborate the singular vector operator and singular value operator in Section 3.2 and Section 3.3, respectively, which forming our core components. The optimization objective is briefly presented in Section 3.4.

### 3.1 OVERVIEW

The intention of the proposed Decomposition Ascribed Synergistic Learning (DASL) is to dedicate the decomposed optimization of degraded singular vectors and singular values respectively, since they naturally undertake the different types of degradation information as observed in Figs. 1 and 2. And the decomposed optimization renders a more unified perspective to revisit diverse degradations for ascribed synergistic learning. Through examining the singular vector dominated degradations which containing *rain, noise, blur*, and singular value dominated degradations including *hazy, low-light*, we make the following assumptions: (i) The singular vectors responsible for the content information and spatial details. (ii) The singular values represent the global statistical properties of the image. Therefore, the optimization of the degraded singular vectors could be performed throughout the backbone network. And the optimization for the degraded singular values can be condensed to a few of pivotal positions. Specifically, we substitute half of the convolution layers with SVEO, which are uniformly distributed across the entire network. While the SVAOs are only performed at the bottleneck layers of the backbone network. We ensure the compatibility between the optimized singular values and singular vectors through remaining regular layers, and the proposed DASL can be lightly integrated into existing image restoration backbone for decomposed optimization.

### 3.2 SINGULAR VECTOR OPERATOR

The singular vector operator is proposed to optimize the degraded singular vectors of the latent representation, and supposed to be decoupled with the optimization of singular values. Explicitly performing the singular value decomposition on high-dimensional tensors solves this problem naturally with little effort, however, the huge computational overhead is non-negligible. Whether can we modify the singular vectors with less computation burden. The answer is affirmative and lies in the orthogonal matrices multiplication.

**Theorem 3.1** *For an arbitrary matrix $X \in \mathbb{R}^{h \times w}$ and random orthogonal matrices $P \in \mathbb{R}^{h \times h}, Q \in \mathbb{R}^{w \times w}$, the products of the $PX$, $XQ$, $PXQ$ have the same singular values with the matrix $X$.*

We provide the proof of theorem 3.1 in the Appendix A.1. In order to construct the orthogonal regularized operator to process the latent representation, the form of the convolution operation

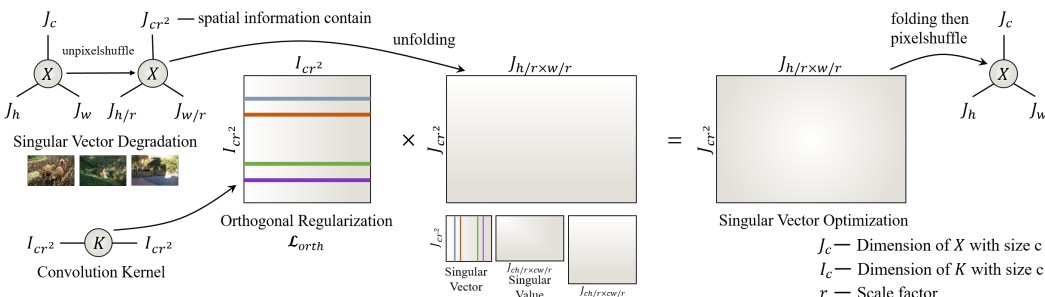

Figure 3: An illustration of the proposed Singular Vector Operator (SVEO), which is dedicated on the optimization of the singular vector dominated degradations, *i.e.*, rain, noise, blur. Theorem 3.1 supports the feasibility and the orthogonal regularization $\mathcal{L}_{orth}$ refers to Eq. 1.

is much eligible than matrix multiplication, which is agnostic to the input resolution. Hence the distinction between these two forms of operation ought to be taken into consideration.

Prior works Sedghi et al. (2019); Jain (1989) have shown that the convolution operation $y = conv(x)$ with kernel size $k \times k$ can be transformed to linear matrix multiplication $vec(y) = A\, vec(x)$. Supposing the processed tensors $y, x \in \mathbb{R}^{1 \times n \times n}$ for simplicity, the size of the projection matrix $A$ will come to be $n^2 \times n^2$ with doubly block circulant, which is intolerable to enforce the orthogonal regularization, especially for high-resolution inputs. Another simple way is to employ the $1 \times 1$ convolution with regularized orthogonality, however, the singular vectors of the latent representation along the channel dimension will be changed rather than spatial dimension.

Inspired by this point, SVEO proposes to transpose spatial information of the latent representation $X \in \mathbb{R}^{c \times h \times w}$ to channel dimension with the ordinary unpixelshuffle operation Shi et al. (2016), resulting in $X' \in \mathbb{R}^{cr^2 \times h/r \times w/r}$. And then applying the orthogonal regularized $1 \times 1$ convolution $\mathcal{K} \in \mathbb{R}^{cr^2 \times cr^2}$ in this domain, as shown in Fig. 3. Thereby, the degraded singular vectors can be revised pertinently, and the common properties among various singular vector dominated degradations can be implicitly exploited. We note that the differences between SVEO and conventional convolution lie in the following: (i) The SVEO is more consistent with the matrix multiplication as it eliminates the overlap operation attached to the convolution. (ii) The weights of SVEO are reduced to matrix instead of tensor, where the orthogonal regularization can be enforced comfortably. Besides, compared to the matrix multiplication, SVEO further utilizes the channel redundancy and spatial adaptivity within a local $r \times r$ region for conducive information utilization. The orthogonal regularization is formulated as

$$\mathcal{L}_{orth} = \|WW^T \odot (\mathbf{1} - I)\|_F^2, \tag{1}$$

where $W$ represents the weight matrix, $\mathbf{1}$ denotes a matrix with all elements set to 1, and $I$ denotes the identity matrix.

### 3.3 SINGULAR VALUE OPERATOR

The singular value operator endeavors to optimize the degraded singular values of the latent representation while supposed to be less entangled with the optimization of singular vectors. However, considering the inherent inaccessibility of the singular values, it is hard to perform the similar operation as SVEO in the same vein. To this end, we instead resort to reconnoitering the essence of singular values and found that it is eminently associated with inverse discrete fourier transform. We provide the formation of a two-dimensional signal represented by singular value decomposition (SVD) and inverse discrete fourier transform (IDFT) in Eq. 2 and Eq. 3 as follows

$$X = U\Sigma V^T = \sum_{i=1}^{k} \sigma_i u_i v_i^T = \sum_{i=1}^{k} \sigma_i X_i, \tag{2}$$

where $X \in \mathbb{R}^{h \times w}$ represents the latent representation and $U \in \mathbb{R}^{h \times h}$, $V \in \mathbb{R}^{w \times w}$ represent the decomposed singular vectors with columns $u_i$, $v_i$, $k = min(h, w)$ denotes the rank of $X$. $\Sigma$ represents the singular values with diagonal elements $\sigma_i$.

$$X = \frac{1}{hw} \sum_{u=0}^{h-1} \sum_{v=0}^{w-1} G(u,v) e^{j2\pi(\frac{um}{h} + \frac{vn}{w})} = \frac{1}{hw} \sum_{u=0}^{h-1} \sum_{v=0}^{w-1} G(u,v)\phi(u,v), \tag{3}$$

Figure 4: An illustration of the core idea of the proposed Singular Value Operator (SVAO), which is dedicated on the optimization of the singular value dominated degradations, *i.e.*, haze and low-light. Two-dimensional signal formations are provided for simplicity.

where $G(u,v)$ denotes the coefficients of the fourier transform of $X$, and $\phi(u,v)$ denotes the corresponding two-dimensional wave component. $m \in \mathbb{R}^{h-1}$, $n \in \mathbb{R}^{w-1}$. Observing that both SVD and IDFT formation can be regarded as a weighted sum on a set of basis, *i.e.*, $u_i v_i^T$ and $e^{j2\pi(\frac{um}{h} + \frac{vn}{w})}$, while the decomposed singular values $\sigma_i$ and the transformed fourier coefficients $G(u,v)$ inherently undertake the same role for the linear combination of various bases.

In Fig. 5, we present the visualized comparison of the reconstruction results using partial components of SVD and IDFT progressively, while both formations conform to the principle from outline to details. Therefore, we presume that the SVD and IDFT operate in a similar way in terms of signal formation, and the combined coefficients $\sigma_i$ and $G(u,v)$ can be approximated to each other.

In this way, we successfully translate the unattainable singular values optimization to the accessible fourier coefficients optimization, as shown in Fig. 4. Considering the decomposed singular values typically characterize the global statistics of the signal, SVAO thus concentrates on the optimization of the norm of $G(u,v)$ for consistency, *i.e.*, the amplitude map, since the phase of $G(u,v)$ implicitly represents the structural

Table 1: Time comparison (ms) between SVD and FFT formation for signal representation on high-dimensional tensor, with supposed size $64 \times 128 \times 128$, where the *Decom.* and *Comp.* represent the decomposition and composition.

| Formation | *Decom.* time | *Comp.* time | Total time |
|---|---|---|---|
| SVD | 180.243 | 0.143 | 180.386 |
| FFT | 0.159 | 0.190 | 0.349 |

content Stark (2013); Oppenheim & Lim (1981) and more in line with the singular vectors. The above two-dimensional signal formation can be easily extended to the three-dimensional tensor to perform the $1 \times 1$ convolution. Since we adopt the SVAO merely in the bottleneck layers of the backbone network with low resolution inputs, and the fast fourier transform is substantially faster than the singular value decomposition; see Table 1. The consequent overhead of SVAO can be greatly compressed. Note that the formation of Eq. 3 is a bit different from the definitive IDFT, and we provide the equivalence proof in the Appendix A.2.

### 3.4 OPTIMIZATION OBJECTIVE

The decomposition loss $\mathcal{L}_{dec}$ is developed to favor the decomposed optimization congruously, formulated as

$$\mathcal{L}_{dec} = \sum_{i=1}^{3} \beta \| U_{rec}^{(i)} V_{rec}^{(i)T} - U_{cle}^{(i)} V_{cle}^{(i)T} \|_1 + \| \Sigma_{rec}^{(i)} - \Sigma_{cle}^{(i)} \|_1, \qquad (4)$$

where $U_{cle}$, $V_{cle}$, and $\Sigma_{cle}$ represent the decomposed singular vectors and singular values of the clean image, $U_{rec}$, $V_{rec}$, and $\Sigma_{rec}$ represent the decomposed singular vectors and singular values of the recovered image. For simplicity, we omit the pseudo-identity matrix between $UV^T$ for dimension transformation. $\beta$ denotes the weight.

The overall optimization objective of DASL comprises the orthogonal regularization loss $\mathcal{L}_{orth}$ and the decomposition loss $\mathcal{L}_{dec}$, together with the original loss functions of the integrated backbone network $\mathcal{L}_{ori}$, formulated as

$$\mathcal{L}_{total} = \mathcal{L}_{ori} + \lambda_{orth} \mathcal{L}_{orth} + \lambda_{dec} \mathcal{L}_{dec}, \qquad (5)$$

where $\lambda_{orth}$ and $\lambda_{dec}$ denote the balanced weights.

*Progressive component reconstruction*

| 5% | 10% | 20% | 40% | 70% | 100% |

Figure 5: Visual comparison of the progressive reconstruction results with SVD and IDFT components, respectively. First row, IDFT reconstruction result. Second row, SVD reconstruction result. Both conform to the principle from outline to details.

Table 2: Quantitative results on five common image restoration datasets with state-of-the-art general image restoration and all-in-one methods. The baseline results are in grey.

| Method | Rain100L PSNR↑ | SSIM↑ | BSD68 PSNR↑ | SSIM↑ | GoPro PSNR↑ | SSIM↑ | SOTS PSNR↑ | SSIM↑ | LOL PSNR↑ | SSIM↑ | Average PSNR↑ | SSIM↑ | Params |
|---|---|---|---|---|---|---|---|---|---|---|---|---|---|
| NAFNet | 35.56 | 0.967 | 31.02 | 0.883 | 26.53 | 0.808 | 25.23 | 0.939 | 20.49 | 0.809 | 27.76 | 0.881 | 17.11M |
| Restormer | 34.81 | 0.962 | 31.49 | 0.884 | 27.22 | 0.829 | 24.09 | 0.927 | 20.41 | 0.806 | 27.60 | 0.881 | 26.13M |
| ShuffleFormer | 35.23 | 0.966 | 31.53 | 0.894 | 27.14 | 0.828 | 24.98 | 0.938 | 20.12 | 0.814 | 27.80 | 0.888 | 50.60M |
| MPRNet | 38.16 | 0.981 | 31.35 | 0.889 | 26.87 | 0.823 | 24.27 | 0.937 | 20.84 | 0.824 | 28.27 | 0.890 | 15.74M |
| DGUNet | 36.62 | 0.971 | 31.10 | 0.883 | 27.25 | 0.837 | 24.78 | 0.940 | 21.87 | 0.823 | 28.32 | 0.891 | 17.33M |
| MambaIR | 34.54 | 0.962 | 31.37 | 0.890 | 26.52 | 0.804 | 25.74 | 0.946 | 18.23 | 0.740 | 27.28 | 0.868 | 1.36M |
| IR-SDE | 35.18 | 0.969 | 30.26 | 0.895 | 25.63 | 0.777 | 24.73 | 0.925 | 11.83 | 0.473 | 25.53 | 0.808 | 137.15M |
| DL | 21.96 | 0.762 | 23.09 | 0.745 | 19.86 | 0.672 | 20.54 | 0.826 | 19.83 | 0.712 | 21.05 | 0.743 | 2.09M |
| Transweather | 29.43 | 0.905 | 29.00 | 0.841 | 25.12 | 0.757 | 21.32 | 0.885 | 21.21 | 0.792 | 25.22 | 0.836 | 37.93M |
| TAPE | 29.67 | 0.904 | 30.18 | 0.855 | 24.47 | 0.763 | 22.16 | 0.861 | 18.97 | 0.621 | 25.09 | 0.801 | 1.07M |
| IDR | 35.63 | 0.965 | 31.60 | 0.887 | 27.87 | 0.846 | 25.24 | 0.943 | 21.34 | 0.826 | 28.34 | 0.893 | 15.34M |
| AirNet | 32.98 | 0.951 | 30.91 | 0.882 | 24.35 | 0.781 | 21.04 | 0.884 | 18.18 | 0.735 | 25.49 | 0.846 | 8.93M |
| PromptIR | 34.24 | 0.957 | 31.30 | 0.885 | 26.43 | 0.802 | 25.18 | 0.934 | 21.69 | 0.805 | 27.76 | 0.876 | 35.59M |
| DA-CLIP | 35.69 | 0.974 | 30.45 | 0.898 | 25.92 | 0.786 | 25.24 | 0.938 | 17.96 | 0.738 | 27.05 | 0.867 | 136.82M |
| DASL+MPRNet | 38.02 | 0.980 | 31.57 | 0.890 | 26.91 | 0.823 | 25.82 | 0.947 | 20.96 | 0.826 | 28.66 | 0.893 | 15.15M |
| DASL+DGUNet | 36.96 | 0.972 | 31.23 | 0.885 | 27.23 | 0.836 | 25.33 | 0.943 | 21.78 | 0.824 | 28.51 | 0.892 | 16.92M |
| DASL+MambaIR | 34.82 | 0.965 | 31.50 | 0.892 | 26.77 | 0.811 | 25.89 | 0.951 | 19.54 | 0.775 | 27.70 | 0.879 | 1.02M |
| DASL+IR-SDE | 35.46 | 0.972 | 30.43 | 0.901 | 25.91 | 0.789 | 25.08 | 0.941 | 15.26 | 0.614 | 26.42 | 0.843 | 128.64M |
| DASL+AirNet | 34.93 | 0.961 | 30.99 | 0.883 | 26.04 | 0.788 | 23.64 | 0.924 | 20.06 | 0.805 | 27.13 | 0.872 | 5.41M |
| DASL+PromptIR | 36.67 | 0.975 | 31.66 | 0.896 | 27.36 | 0.839 | 25.55 | 0.944 | 21.73 | 0.834 | 28.59 | 0.897 | 32.31M |
| DASL+DA-CLIP | 35.78 | 0.979 | 30.87 | 0.901 | 26.08 | 0.789 | 25.53 | 0.947 | 19.21 | 0.753 | 27.49 | 0.874 | 130.45M |

## 4 EXPERIMENTS

In this section, we first clarify the experimental settings, and then present the qualitative and quantitative comparison results with eleven baseline methods for unified image restoration. Moreover, extensive ablation experiments are conducted to verify the effectiveness of our method.

### 4.1 IMPLEMENTATION DETAILS

**Tasks and Metrics.** We train our method on five image restoration tasks synchronously. The corresponding training set includes Rain200L Yang et al. (2017) for image deraining, RESIDE-OTS Li et al. (2018) for image dehazing, BSD400 Martin et al. (2001), WED Ma et al. (2016) for image denoising, GoPro Nah et al. (2017) for image deblurring, and LOL Chen et al. (2018) for low-light image enhancement. For evaluation, 100 image pairs in Rain100L Yang et al. (2017), 500 image pairs in SOTS-Outdoor Li et al. (2018), total 192 images in CBSD68 Martin et al. (2001), Urban100 Huang et al. (2015) and Kodak24 Franzen (1999), 1111 image pairs in GoPro Nah et al. (2017), 15 image pairs in LOL Chen et al. (2018) are utilized as the test set. We report the Peak Signal to Noise Ratio (PSNR) and Structural Similarity (SSIM) as numerical metrics.

**Training.** We implement our method on single NVIDIA Geforce RTX 3090 GPU. For fair comparison, all comparison methods have been retrained in the new mixed dataset with their default hyper parameter settings. We adopt the MPRNet Zamir et al. (2021), DGUNet Mou et al. (2022), and AirNet Li et al. (2022) as our baseline to validate the proposed Decomposition Ascribed Synergistic Learning. The entire network is trained with Adam optimizer for 1200 epochs. We set the batch size as 8 and random crop 128x128 patch from the original image as network input after data augmentation. We set the $\beta$ in $\mathcal{L}_{dec}$ as 0.01, and the $\lambda_{orth}$, $\lambda_{dec}$ are set to be 1e-4 and 0.1, respectively. We perform evaluations every 20 epochs with the highest average PSNR scores as the final parameters result. More model details and training protocols are presented in the Appendix B.

Table 3: Quantitative results of image denoising on CBSD68, Urban100 and Kodak24 datasets (PSNR↑).

| Method | CBSD68 | | | Urban100 | | | Kodak24 | | |
|---|---|---|---|---|---|---|---|---|---|
| | $\sigma=15$ | $\sigma=25$ | $\sigma=50$ | $\sigma=15$ | $\sigma=25$ | $\sigma=50$ | $\sigma=15$ | $\sigma=25$ | $\sigma=50$ |
| NAFNet | 33.67 | 31.02 | 27.73 | 33.14 | 30.64 | 27.20 | 34.27 | 31.80 | 28.62 |
| Restormer | 34.03 | 31.49 | 28.11 | 33.72 | 31.26 | 28.03 | 34.78 | 32.37 | 29.08 |
| MPRNet | 34.01 | 31.35 | 28.08 | 34.13 | 31.75 | 28.41 | 34.77 | 32.31 | 29.11 |
| DGUNet | 33.85 | 31.10 | 27.92 | 33.67 | 31.27 | 27.94 | 34.56 | 32.10 | 28.91 |
| MambaIR | 33.99 | 31.37 | 28.12 | 34.16 | 31.83 | 28.43 | 34.78 | 32.38 | 29.10 |
| IR-SDE | 33.23 | 30.26 | 26.92 | 32.31 | 29.85 | 26.93 | 33.82 | 31.08 | 27.79 |
| DL | 23.16 | 23.09 | 22.09 | 21.10 | 21.28 | 20.42 | 22.63 | 22.66 | 21.95 |
| Transweather | 31.16 | 29.00 | 26.08 | 29.64 | 27.97 | 26.08 | 31.67 | 29.64 | 26.74 |
| TAPE | 32.86 | 30.18 | 26.63 | 32.19 | 29.65 | 25.87 | 33.24 | 30.70 | 27.19 |
| AirNet | 33.49 | 30.91 | 27.66 | 33.16 | 30.83 | 27.45 | 34.14 | 31.74 | 28.59 |
| PromptIR | 33.92 | 31.30 | 28.02 | 34.06 | 31.69 | 28.38 | 34.63 | 32.19 | 29.04 |
| DA-CLIP | 33.35 | 30.45 | 27.14 | 32.83 | 30.24 | 27.29 | 34.03 | 31.26 | 27.98 |
| DASL+MPRNet | 34.16 | 31.57 | 28.18 | 34.21 | 31.82 | 28.47 | 34.91 | 32.46 | 29.18 |
| DASL+DGUNet | 33.94 | 31.23 | 27.94 | 33.74 | 31.31 | 27.96 | 34.69 | 32.16 | 28.93 |
| DASL+MambaIR | 34.12 | 31.50 | 28.27 | 34.31 | 32.04 | 28.61 | 35.04 | 32.64 | 29.36 |
| DASL+IR-SDE | 33.38 | 30.43 | 27.09 | 32.42 | 29.97 | 27.05 | 34.01 | 31.26 | 27.95 |
| DASL+AirNet | 33.69 | 30.99 | 27.68 | 33.35 | 30.89 | 27.46 | 34.32 | 31.79 | 28.61 |
| DASL+PromptIR | 34.24 | 31.66 | 28.33 | 34.20 | 31.84 | 28.51 | 34.94 | 32.41 | 29.36 |
| DASL+DA-CLIP | 33.70 | 30.87 | 27.55 | 33.03 | 30.47 | 27.50 | 34.46 | 31.67 | 28.38 |

Table 4: Evaluating the scalability of decomposed optimization on the full set with merely trained on singular vector dominated degradations (*vec.*) and singular value dominated degradations (*val.*) (PSNR↑).

| Tasks | Rain100L | BSD68 | GoPro | SOTS | LOL |
|---|---|---|---|---|---|
| MPRNet (*vec.*) | 39.47 | 31.50 | 27.61 | 15.91 | 7.77 |
| MPRNet (*val.*) | - | - | - | - | - |
| DGUNet (*vec.*) | 39.04 | 31.46 | 28.22 | 15.92 | 7.76 |
| DGUNet (*val.*) | 23.10 | 20.39 | 21.84 | 24.59 | 20.45 |
| MambaIR (*vec.*) | 37.21 | 31.53 | 27.86 | 16.63 | 7.76 |
| MambaIR (*val.*) | 21.65 | 20.52 | 19.67 | 25.62 | 21.16 |
| AirNet (*vec.*) | 36.62 | 31.33 | 26.35 | 15.90 | 7.75 |
| AirNet (*val.*) | 19.52 | 19.1 | 14.47 | 20.63 | 16.01 |
| DASL+MPRNet (*vec.*) | 39.39 | 31.63 | 27.57 | 17.21 | 11.23 |
| DASL+MPRNet (*val.*) | 21.87 | 19.96 | 21.35 | 25.13 | 20.33 |
| DASL+DGUNet (*vec.*) | 39.11 | 31.55 | 28.16 | 16.87 | 10.21 |
| DASL+DGUNet (*val.*) | 23.19 | 20.28 | 22.69 | 25.05 | 20.87 |
| DASL+MambaIR (*vec.*) | 37.44 | 31.59 | 27.92 | 16.96 | 9.32 |
| DASL+MambaIR (*val.*) | 21.78 | 20.91 | 20.35 | 25.79 | 21.82 |
| DASL+AirNet (*vec.*) | 36.87 | 31.22 | 26.72 | 15.97 | 8.77 |
| DASL+AirNet (*val.*) | 21.25 | 20.38 | 21.12 | 24.60 | 20.58 |

## 4.2 COMPARISON WITH STATE-OF-THE-ART METHODS

We compare our DASL with comprehensive state-of-the-art methods, including general image restoration methods: NAFNet Chen et al. (2022a), Restormer Zamir et al. (2022b), ShuffleFormer Xiao et al. (2023), MPRNet Zamir et al. (2021), DGUNet Mou et al. (2022),MambaIR Guo et al. (2024), IR-SDE Luo et al. (2023b), and all-in-one fashion methods: DL Fan et al. (2019), Transweather Valanarasu et al. (2022), TAPE Liu et al. (2022), AirNet Li et al. (2022), IDR Zhang et al. (2023), PromptIR Potlapalli et al. (2024), and DA-CLIP Luo et al. (2023a) on five image restoration tasks.

Table 2 reports the quantitative comparison results. It can be observed that the performance of the general image restoration methods is systematically superior to the professional all-in-one methods when more degradations are involved, attributed to the large model size. While our DASL further advances the backbone network capability with fewer parameters, owing to the implicit synergistic learning. We provide more visual comparison results of the DASL integration against the vanilla baselines in Appendix H. Consistent with existing unified image restoration methods Zamir et al. (2022b); Li et al. (2022), we report the detailed denoising results at different noise ratio in Table 3, where the performance gain are consistent.

In Table 5, we present the computation overhead involved in DASL, where the FLOPs and inference time are calculated over 100 testing images with the size of 512×512. It can be observed that our DASL substantially reduces the computation complexity of the baseline methods with considerable inference acceleration, *e.g.* 12.86% accelerated on MPRNet and 58.61% accelerated on Air-

Table 5: Comparison of the model size and computation complexity between baseline / DASL.

| Method | Params (M) | FLOPs (B) | Inference Time (s) |
|---|---|---|---|
| MambaIR | 1.36 / 1.02 | 224.07 / 196.72 | 1.184 / 1.126 |
| IR-SDE | 137.15 / 128.64 | 1517.34 / 1386.27 | 18.10 / 17.21 |
| MPRNet | 15.74 / 15.15 | 5575.32 / 2905.14 | 0.241 / 0.210 |
| DGUNet | 17.33 / 16.92 | 3463.66 / 3020.22 | 0.397 / 0.391 |
| AirNet | 8.93 / 5.41 | 1205.09 / 767.89 | 0.459 / 0.190 |

Net. We present the bountiful visual comparison results in the Appendix H, while our DASL exhibits superior visual recovery quality, *i.e.*, more precise details in singular vector dominated degradations and more stable global recovery in singular value dominated degradations.

## 4.3 ABLATION STUDIES

We present the ablation experiments on the combined degradation dataset with MPRNet as the backbone to verify the effectiveness of our method. In Table 6, we quantitatively evaluate the two developed operators SVEO and SVAO, and the decomposition loss. The metrics are reported on the each of degradations in detail, from which we can make the following observations: **a)** Both SVEO and SVAO are beneficial for advancing the unified degradation restoration performance, attributing to the ascribed synergistic learning. **b)** The congruous decomposition loss is capable to work alone,

Table 6: Ablation experiments on the components design.

| Method | SVEO | SVAO | $\mathcal{L}_{orth}$ | $\mathcal{L}_{dec}$ | Rain100L PSNR↑ | SSIM↑ | BSD68 PSNR↑ | SSIM↑ | GoPro PSNR↑ | SSIM↑ | SOTS PSNR↑ | SSIM↑ | LOL PSNR↑ | SSIM↑ | Avg. PSNR↑ | SSIM↑ |
|---|---|---|---|---|---|---|---|---|---|---|---|---|---|---|---|---|
| Baseline | | | | | 38.16 | 0.981 | 31.35 | 0.889 | 26.87 | 0.823 | 24.27 | 0.937 | 20.84 | 0.824 | 28.27 | 0.890 |
| With no orth. *SVEO* | ✓ | | | | 37.73 | 0.981 | 31.31 | 0.889 | 26.79 | 0.819 | 24.63 | 0.939 | 20.83 | 0.824 | 28.26 | 0.890 |
| With *SVAO* | | ✓ | | | 37.92 | 0.980 | 31.41 | 0.889 | 26.85 | 0.821 | 25.58 | 0.943 | 21.05 | 0.828 | 28.56 | 0.892 |
| With *SVEO* | ✓ | | ✓ | | 38.04 | 0.981 | 31.46 | 0.890 | 26.97 | 0.826 | 25.53 | 0.945 | 20.76 | 0.822 | 28.55 | 0.893 |
| With *SVEO* and *SVAO* | ✓ | ✓ | ✓ | | 38.01 | 0.980 | 31.53 | 0.890 | 26.94 | 0.825 | 25.63 | 0.948 | 20.92 | 0.826 | 28.61 | 0.893 |
| With $\mathcal{L}_{dec}$ | | | | ✓ | 38.10 | 0.982 | 31.39 | 0.889 | 26.78 | 0.819 | 24.70 | 0.942 | 20.98 | 0.827 | 28.39 | 0.892 |
| DASL+MPRNet | ✓ | ✓ | ✓ | ✓ | 38.02 | 0.980 | 31.57 | 0.890 | 26.91 | 0.823 | 25.82 | 0.947 | 20.96 | 0.826 | 28.66 | 0.893 |

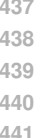
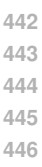
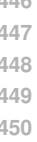
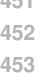
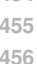

Figure 6: Evaluating the synergy effect through training trajectory between baseline and DASL on *vec.* dominated degradations.

Figure 7: Evaluating the synergy effect through training trajectory between baseline and DASL on *val.* dominated degradations.

and well collaborated with developed operators for decomposed optimization. **c)** The orthogonal regularization is crucial to the reliable optimization of SVEO for preventing the performance drop.

To further verify the scalability of the decomposed optimization, Table 4 evaluates the performance with partially trained on singular vector dominated degradations (*vec.*) and singular value dominated degradations (*val.*). While some properties have been observed: **a)** Basically, the baseline methods concentrate on the trainable degradations, while our DASL further contemplates the untrainable ones in virtue of its slight task dependency. **b)** The performance of MPRNet on *val.* is unattainable due to the non-convergence, however, our DASL successfully circumvents this drawback owing to the more unified decomposed optimization on singular values rather than task-level learning. **c)** The *vec.* seems to be supportive to the restoration performance of *val.*, see the comparison of Tables 2 and 4, indicating the potential relationship among decomposed two types of degradations.

We present the comparison of the training trajectory between baseline and DASL on singular vector dominated and singular value dominated degradations in Figs. 6 and 7. It can be observed that our DASL significantly suppresses the drastic optimization process, retaining the overall steady to better convergence point with even fewer parameters, attributing to the ascribed synergistic learning.

## 4.4 LATENT SPACE ANALYSIS

The decomposition ascribed degradation analysis has been clearly unveiled in pixel space so far, however, whether the property can be generalized to latent space is more appealing, which is exactly what the DASL built upon for synergestic optimization. In Tab. 7, we provide the validation for latent degradation analysis, where we train a linear projector to transform the degraded latents to clean latents and compute the reconstruction error and transmission ratio Shi et al. (2024). The clean latents are obtained with clean input and extracted from the same layer as degraded latents. To see how well the singular vectors and singular values represent the degradation information, two

Table 7: Validation of the decomposition ascribed degradation analysis in latent space.

| Source | Rain100L Loss | Ratio | BSD68 Loss | Ratio | GoPro Loss | Ratio | SOTS Loss | Ratio | LOL Loss | Ratio |
|---|---|---|---|---|---|---|---|---|---|---|
| Degraded | 0.00107 | 66.1% | 0.00143 | 63.7% | 0.00134 | 55.5% | 0.00251 | 50.1% | 0.00943 | 15.4% |
| Swap *Vec.* | 0.00034 | **94.3%** | 0.00043 | **92.2%** | 0.00042 | **81.9%** | 0.00219 | 53.1% | 0.00926 | 16.1% |
| Swap *Val.* | 0.00091 | 70.1% | 0.00106 | 70.8% | 0.00129 | 56.3% | 0.00076 | **76.8%** | 0.00118 | **71.2%** |
| Clean | 0.00027 | 100% | 0.00031 | 100% | 0.00019 | 100% | 0.00027 | 100% | 0.00039 | 100% |

variants are conducted with firstly swapping clean singular vectors or singular values to degraded latents, and then perform the linear transformation. The singular vector dominated degradations show noticeable transmission ratio improvement ($\sim$30%) when swapped with clean singular vectors, and exhibit minor transmission ratio improvement ($\sim$5%) when swapped with clean singular values. The same thing also happened in singular value dominated degradations with $\sim$ 25%-50% transmission improvement when swapped with clean singular values and $\sim$3% transmission improvement when swapped with clean singular vectors. Therefore, we have reason to suggest that the latent degradation analysis is consistency with pixel-space degradation analysis in ascribing degradation types.

## 5 CONCLUSION

In this paper, we revisited the diverse degradations through the lens of singular value decomposition and observed that the decomposed singular vectors and singular values naturally undertake the different types of degradation information, ascribing various restoration tasks into two groups, *i.e.*, singular vector dominated degradations and singular value dominated degradations. The proposed Decomposition Ascribed Synergistic Learning dedicates the decomposed optimization of degraded singular vectors and singular values respectively, rendering a more unified perspective to inherently utilize the potential partnership among diverse restoration tasks for ascribed synergistic learning. Furthermore, two effective operators SVEO and SVAO have been developed to favor the decomposed optimization, along with a congruous decomposition loss, which can be lightly integrated into existing image restoration backbone. Extensive experiments on bunch of image restoration tasks validated the effectiveness of the proposed method and the generality of the SVD-based degradation analysis.

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

## A    PROOF OF THE PROPOSITIONS

### A.1    PROOF OF THEOREM 1

**Theorem A.1** *For an arbitrary matrix $X \in \mathbb{R}^{h \times w}$ and random orthogonal matrices $P \in \mathbb{R}^{h \times h}, Q \in \mathbb{R}^{w \times w}$, the products of the $PX$, $XQ$, $PXQ$ have the same singular values with the matrix $X$.*

*Proof.* According to the definition of Singular Value Decomposition (SVD), we can decompose matrix $X \in \mathbb{R}^{h \times w}$ into $USV^T$, where $U \in \mathbb{R}^{h \times h}$ and $V \in \mathbb{R}^{w \times w}$ indicate the orthogonal singular vector matrices, $S \in \mathbb{R}^{h \times w}$ indicates the diagonal singular value matrix. Thus $X^{'} = PXQ = PUSV^TQ$. Denotes $U^{'} = PU$ and $V^{'T} = V^TQ$, then $X^{'}$ can be decomposed into $U^{'}SV^{'T}$ if $U^{'}$ and $V^{'T}$ are orthogonal matrices.

$$U^{'-1} = (PU)^{-1} = U^{-1}P^{-1} = U^TP^T = (PU)^T = U^{'T} \tag{6}$$

$$(V^{'T})^{-1} = (V^TQ)^{-1} = Q^{-1}(V^T)^{-1} = Q^TV = (V^TQ)^T = V^{'} \tag{7}$$

Therefore, $U^{'}U^{'T} = I$ and $V^{'T}V^{'} = I$, where $I$ denotes the identity matrix, and $U^{'}$, $V^{'T}$ are orthogonal. $X^{'}$ and $X$ have the same singular values $S$, and the singular vectors of $X$ can be orthogonally transformed to $PU$, $Q^TV$. Correspondingly, it can be easily extended to the case of $PX$ and $XQ$. $\qquad\square$

### A.2    EQUIVALENCE PROOF OF EQUATION 3 AND IDFT

**Proposition.** *The signal formation principle in Equation 3 is equivalence to the definitive Inverse Discrete Fourier Transform (IDFT), where we restate the Equation 3 as following:*

$$X = \frac{1}{hw} \sum_{u=0}^{h-1} \sum_{v=0}^{w-1} G(u,v) e^{j2\pi(\frac{um}{h} + \frac{vn}{w})}, \ m \in \mathbb{R}^{h-1}, n \in \mathbb{R}^{w-1}. \tag{8}$$

*Proof.* For the two-dimensional signal $X \in \mathbb{R}^{h \times w}$, we can represent any point on it through IDFT. Supposing $(m,n)$ and $(m^{'}, n^{'})$ are two random points on $X$, where $m, m^{'} \in [0, h\text{-}1]$, $n, n^{'} \in [0, w\text{-}1]$, and $(m,n) \neq (m^{'}, n^{'})$, we have

$$X(m,n) = \frac{1}{hw} \sum_{u=0}^{h-1} \sum_{v=0}^{w-1} G(u,v) e^{j2\pi(\frac{um}{h} + \frac{vn}{w})}, \tag{9}$$

$$X(m^{'}, n^{'}) = \frac{1}{hw} \sum_{u=0}^{h-1} \sum_{v=0}^{w-1} G(u,v) e^{j2\pi(\frac{um^{'}}{h} + \frac{vn^{'}}{w})}. \tag{10}$$

$X(m,n)$ represents the signal value at $(m,n)$ position on $X$, and the same as $X(m^{'}, n^{'})$. Thus, we can rewrite $X$ as

$$\begin{aligned} X &= \begin{bmatrix} X(0,0) & \cdots & X(0, w-1) \\ \vdots & \ddots & \vdots \\ X(h-1, 0) & \cdots & X(h-1, w-1) \end{bmatrix} \\ &= \frac{1}{hw} \sum_{u=0}^{h-1} \sum_{v=0}^{w-1} G(u,v) \cdot \begin{bmatrix} e^{j2\pi(\frac{u0}{h} + \frac{v0}{w})} & \cdots & e^{j2\pi(\frac{u0}{h} + \frac{v(w-1)}{w})} \\ \vdots & \ddots & \vdots \\ e^{j2\pi(\frac{u(h-1)}{h} + \frac{v0}{w})} & \cdots & e^{j2\pi(\frac{u(h-1)}{h} + \frac{v(w-1)}{w})} \end{bmatrix} \\ &= \frac{1}{hw} \sum_{u=0}^{h-1} \sum_{v=0}^{w-1} G(u,v) e^{j2\pi(\frac{um}{h} + \frac{vn}{w})}, \end{aligned} \tag{11}$$

where $m \in \mathbb{R}^{h-1}, n \in \mathbb{R}^{w-1}$. And the two-dimensional wave $e^{j2\pi(\frac{um}{h} + \frac{vn}{w})} \in \mathbb{R}^{h-1 \times w-1}$ denotes the base component. Therefore, the formation principle of Eq 3 is equivalent to the definitive IDFT, *i.e.*, Eqs 9 and 10. $\square$

## B  MODEL DETAILS AND TRAINING PROTOCOLS

We implement our DASL with integrated MPRNet Zamir et al. (2021), DGUNet Mou et al. (2022), and AirNet Li et al. (2022) backbone to validate the effectiveness of the decomposed optimization. All experiments are conducted using PyTorch, with model details and training protocols provided in Table 8. Fig. 8 (a) presents the compound working flow of our operator. Note that the SVAO is only adopted in the bottleneck layer, as described in Section 3.1. We introduce how we embed our operator into the backbone network from a microscopic perspective. Sincerely, the most convenient way is to directly reform the basic block of the backbone network. We present two fashions of the basic block of baseline in Fig. 8 (b) and (c), where the MPRNet fashion is composed of two basic units, *e.g.*, channel attention block (CAB) Zhang et al. (2018), and DGUNet is constructed by two vanilla activated convolutions. We simply replace one of them (dashed line) with our operator to realize the DASL integration. Note that AirNet shares the similar fashion as MPRNet.

Table 8: Model details and training protocols for DASL integrated baselines.

| Configurations | MPRNet | DGUNet | AirNet |
|---|---|---|---|
| optimizer | Adam | Adam | Adam |
| base learning rate | 2e-4 | 1e-4 | 1e-3 |
| learning rate schedular | Cosine decay | Cosine decay | Linear decay |
| momentum of Adam | $\beta_1 = 0.9, \beta_2 = 0.999$ | $\beta_1 = 0.9, \beta_2 = 0.999$ | $\beta_1 = 0.9, \beta_2 = 0.99$ |
| channel dimension | 80 | 80 | 256 |
| augmentation | RandomCropFlip | RandomCropFlip | RandomCropFlip |
| *num.* of replaced operator | 18 | 14 | 50 |
| basic block | channel attention block | activated convolution | degradation guided module |
| optimization objective | CharbonnierLoss + EdgeLoss | CharbonnierLoss + EdgeLoss | L1Loss |

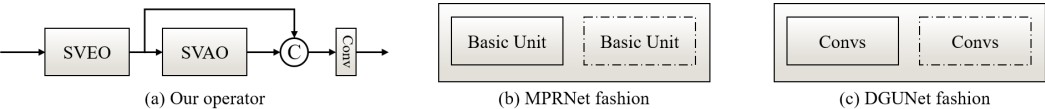

(a) Our operator     (b) MPRNet fashion     (c) DGUNet fashion

Figure 8: The strategy of model integration with DASL. (a) The working flow of our operator. (b) The basic building block of MPRNet fashion. (c) The basic building block of DGUNet fashion.

## C  TRIVIAL ABLATIONS ON OPERATOR DESIGN

The ablation experiment on the choice of scale factor r in SVEO is provided in Table 9. Note that the larger r will incur larger model size. We empirically set the scale ratio r in SVEO as 2. The working flow ablation of combined operator is provided in Table 10, and the compound fashion is preferred.

Table 9: Ablation experiments on the scale ratio $r$ in the SVEO (PSNR↑).

Table 10: Ablation experiments on the working flow of the combined operator (PSNR↑).

| Scale ratio | Rain100L | SOTS | GoPro | BSD68 | LOL |
|---|---|---|---|---|---|
| 1 | 38.01 | 31.55 | 26.88 | 25.84 | 20.93 |
| 2 | 38.02 | 31.57 | 26.91 | 25.82 | 20.96 |
| 4 | 38.07 | 31.58 | 26.92 | 25.81 | 20.98 |

| Working flow | Rain100L | SOTS | GoPro | BSD68 | LOL |
|---|---|---|---|---|---|
| cascaded | 38.01 | 31.55 | 26.88 | 25.84 | 20.87 |
| parallel | 38.02 | 31.57 | 26.91 | 25.82 | 20.88 |
| cascaded + parallel | 38.02 | 31.57 | 26.91 | 25.82 | 20.96 |

## D  EXTENSION EXPERIMENTS FOR PROPERTY VALIDATION

In Table 11, we provide the performance of DASL on real-world image restoration tasks, *i.e.*, under-display camera (UDC) image enhancement. Typically, images captured under UDC system suffer from both blurring due to the spread point spread function, and lower light transmission rate.

Compared to vanilla baseline models, DASL is capable of boosting the performance consistently. Note that the above experiments are performed on real-world UDC dataset Zhou et al. (2021b) without any fine-tuning, validating the capability of the model for processing undesirable degradations. Table 12 evaluates the potential of DASL integration on transformer-based image restoration backbone. Albeit the convolutional form of the developed decomposed operators, the supposed architecture incompatibility problem is not come to be an obstacle. Note that we replace the projection layer at the end of the attention mechanism with developed operators for transformer-based methods.

Table 11: Quantitative results of real-world image restoration tasks (under-display camera image enhancement) on TOLED and POLED datasets.

| Method | TOLED | | | POLED | | |
|---|---|---|---|---|---|---|
| | PSNR↑ | SSIM↑ | LPIPS↓ | PSNR↑ | SSIM↑ | LPIPS↓ |
| MPRNet | 24.69 | 0.707 | 0.347 | 8.34 | 0.365 | 0.798 |
| DGUNet | 19.67 | 0.627 | 0.384 | 8.88 | 0.391 | 0.810 |
| AirNet | 14.58 | 0.609 | 0.445 | 7.53 | 0.350 | 0.820 |
| DASL+MPRNet | 25.65 | 0.733 | 0.326 | 8.95 | 0.392 | 0.788 |
| DASL+DGUNet | 25.25 | 0.727 | 0.329 | 9.80 | 0.410 | 0.783 |
| DASL+AirNet | 18.83 | 0.637 | 0.426 | 9.13 | 0.398 | 0.784 |

Table 12: Evaluating the generality of the DASL integration on transformer-based image restoration backbone among five common image restoration tasks (PSNR↑).

| Methods | Rain100L | BSD68 | GoPro | SOTS | LOL |
|---|---|---|---|---|---|
| SwinIR | 30.78 | 30.59 | 24.52 | 21.50 | 17.81 |
| Restormer | 34.81 | 31.49 | 27.22 | 24.09 | 20.41 |
| ShuffleFormer | 35.23 | 31.53 | 27.14 | 24.98 | 20.12 |
| DASL+SwinIR | 33.53 | 30.84 | 25.72 | 24.10 | 20.36 |
| DASL+Restormer | 35.79 | 31.67 | 27.35 | 25.90 | 21.39 |
| DASL+ShuffleFormer | 35.92 | 31.59 | 27.44 | 25.08 | 20.18 |

# E    CULTIVATING THE SVD POTENTIAL FOR IMAGE RESTORATION.

In fact, Singular Value Decomposition (SVD) has been widely applied for a range of image restoration tasks, such as image denoising, image compression, etc., attributing to the attractive rank properties Sadek (2012) including *truncated energy maximization* and *orthogonal subspaces projection*. The former takes the fact that SVD provides the optima low rank approximation of the signal in terms of dominant energy preservation, which could greatly benefit the signal compression. The latter exploits the fact that the separate order of SVD-decomposed components are orthogonal, which inherently partition the signal into independent rank space, e.g., signal and noise space or range and null space for further manipulation, supporting the application of image denoising or even prevailing inverse problem solvers Wang et al. (2022b). Moreover, the SVD-based degradation analysis proposed in this work excavates another promising property of SVD from the vector-value perspective, which is essentially different from previous rank-based method. Encouragingly, the above two perspectives have the potential to collaborate well and the separate order property is supposed to be incorporated into the DASL for sophisticated degradation relationship investigation in future works. We note that the above two SVD perspectives have the opportunity to collaborate well and the separate order potential is supposed to be incorporated into the DASL for sophisticated relationship investigation in future works.

# F    BROADER IMPACT

This work potentially release the redundant model deployments in real world scenarios, and sincerely benefits a lot of edge applications with limited resources, such as mobile photography and 24/7 surveillance. The privacy of our method may raise potential concerns when considering the removal of some important occlusions in the original images, resulting in the disclosure of private information. Therefore, how to ensure the user-agnostic security of our method needs further investment.

# G    MORE DEGRADATION ANALYSIS AND GENERALIZABLE VERIFICATION

We provide more visual results of decomposition ascribed analysis for diverse degradations in Figs. 11 and 12, to further verify our observation that the decomposed singular vectors and singular values naturally undertake the different types of degradation information. In Figs. 10 and 13, we provide more degradation analysis to validate the generality of the proposed decomposition ascribed analysis, including *downsampling, compression, color shifting, underwater enhancement, and sandstorm enhancement*. The former three types are ascribed into singular vector dominated degradations and the latter two types are ascribed into singular value dominated degradations.

**Experimentally**, if we reexamine the two groups of degradation through SVD-ascribed analysis, namely, *rain, noise, blur, downsampling, compression, color shifting* in singular vector dominated and *hazy, low-light, underwater enhancement, sandstorm enhancement* in singular value dominated,

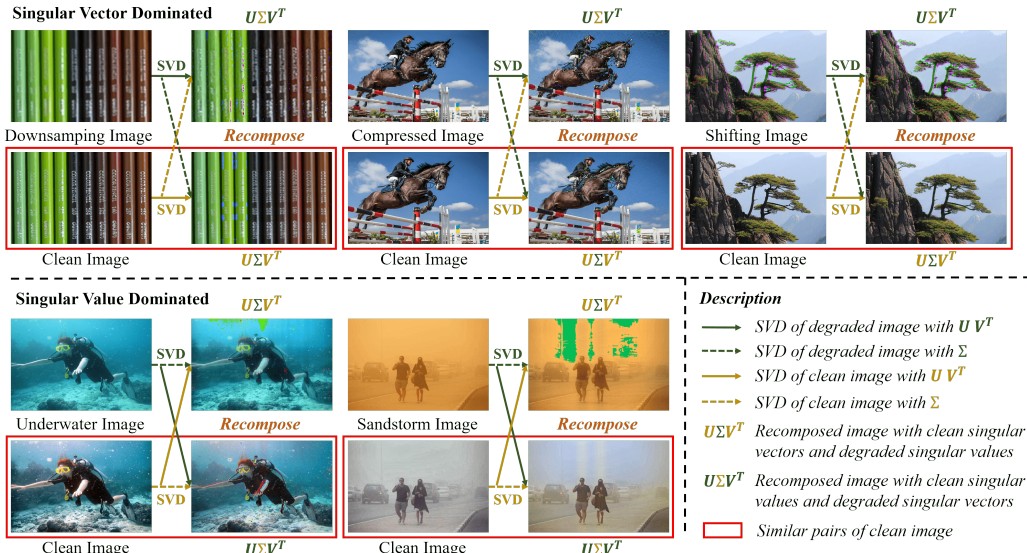

Figure 9: An illustration of the decomposition ascribed degradation analysis on various image restoration tasks through the lens of the singular value decomposition.

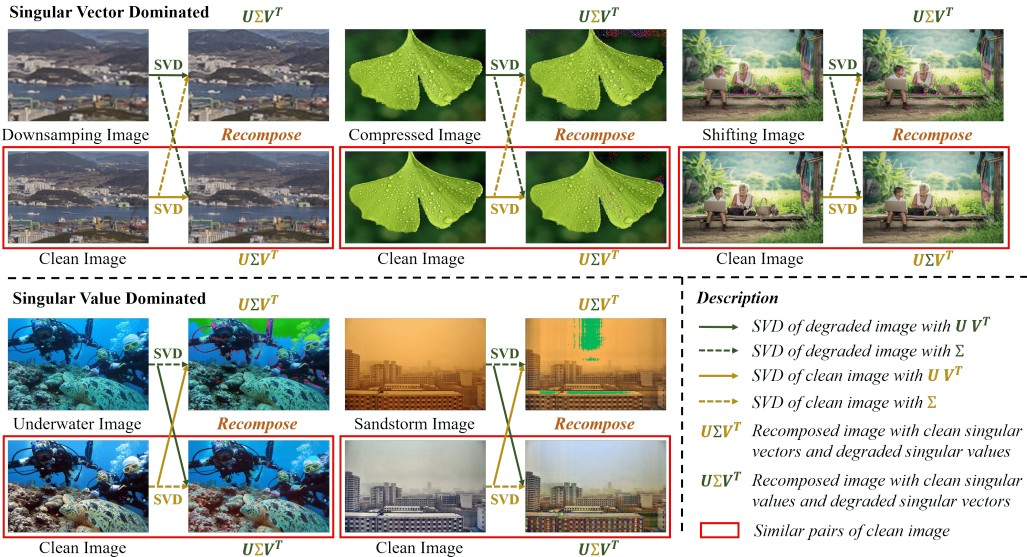

Figure 10: An illustration of the decomposition ascribed degradation analysis on various image restoration tasks through the lens of the singular value decomposition.

it can be concluded that the singular vectors responsible for the spatial content information, while the singular values represent the statistical properties of the image.

**Theoretically**, we verify the above conjecture from the signal formation perspective of SVD, where any signal can be regarded as a weighted sum on a set of basis, i.e., $X = U\Sigma V^T = \sum_{i=1}^{k} \sigma_i u_i v_i^T$. In this light, the singular vectors $\cup_{i=1}^{k}\{u_i v_i^T\}$ represent the base components of the signal for content composition, and the singular values $\cup_{i=1}^{k}\{\sigma_i\}$ represent the combined coefficients for statistical modulation. Their respective dominated degradation types are fundamentally determined by such signal formation properties, i.e., content corruption and statistic distortion. Owing to the closed form of the signal formation principle of SVD, the decomposition ascribed degradation analysis is theoretically generalizable to most of scenes.

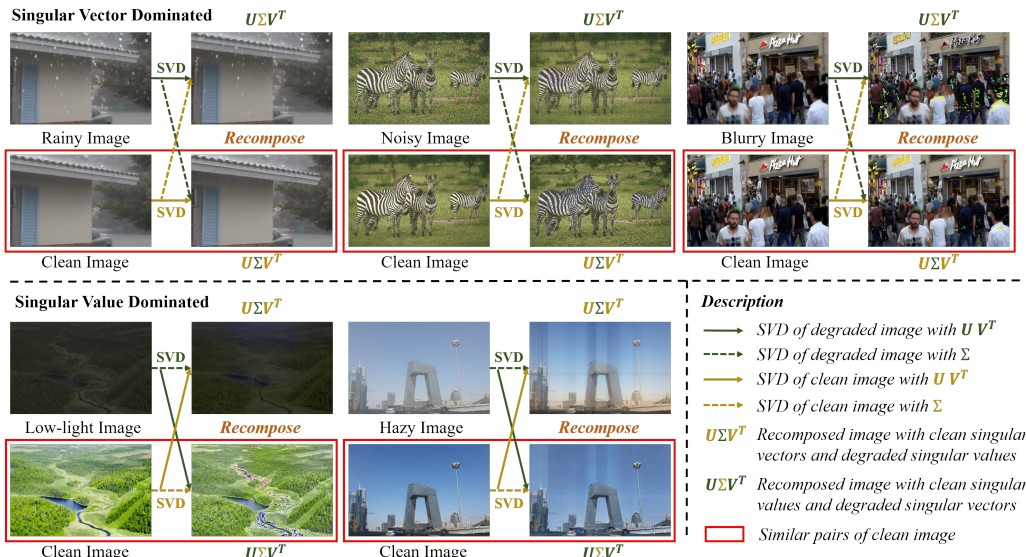

Figure 11: An illustration of the decomposition ascribed degradation analysis on various image restoration tasks through the lens of the singular value decomposition.

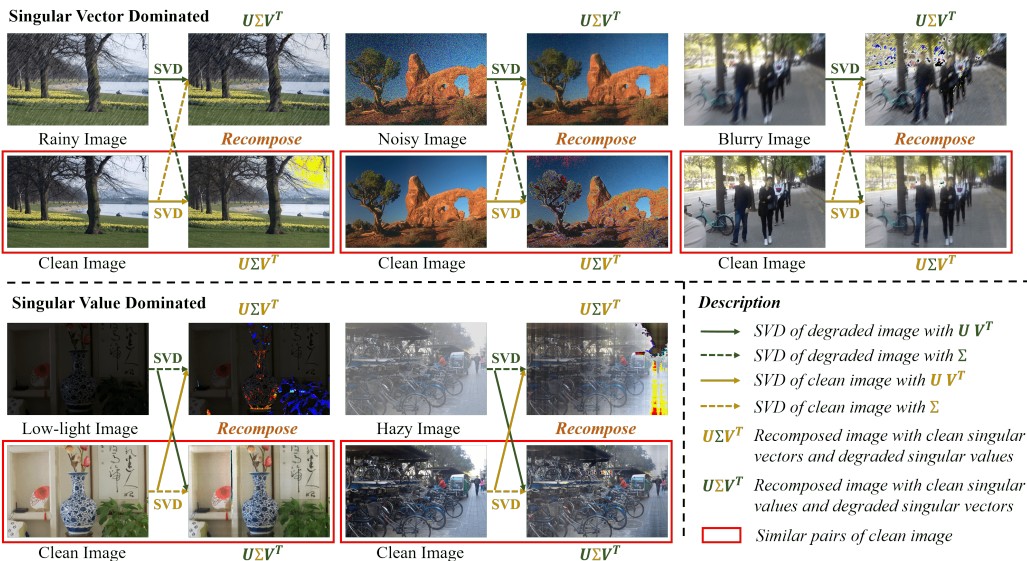

Figure 12: An illustration of the decomposition ascribed degradation analysis on various image restoration tasks through the lens of the singular value decomposition.

## H    VISUAL COMPARISON RESULTS

We present the visual comparison results of the aforementioned image restoration tasks in Figs. 14 to 18, including singular vector dominated degradations *rain*, *noise*, *blur*, and singular value dominated degradations *low-light*, *haze*. It can be observed that our DASL exhibits superior visual recovery quality in both types of degradation, *i.e.*, more precise content details in singular vector dominated degradations and more stable global recovery in singular value dominated degradations, compared to the integrated baseline method.

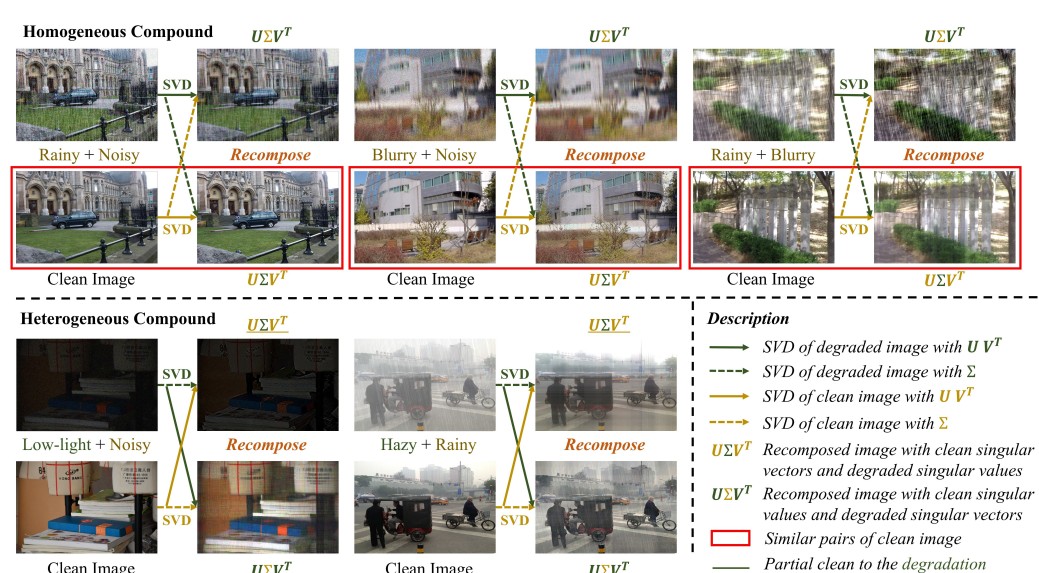

Figure 13: An illustration of the decomposition ascribed degradation analysis on compound degradations, where the pre-ascribed singular vector dominated degradations and singular value dominated degradations are marked. The homogeneous compound degradations refer to that both degradations are ascribed at one side of singular vectors or singular values, and the heterogeneous compound degradations refer to that the degradations are ascribed at both side of singular vectors and singular values, and is consistent with ascription in single degradation analysis.

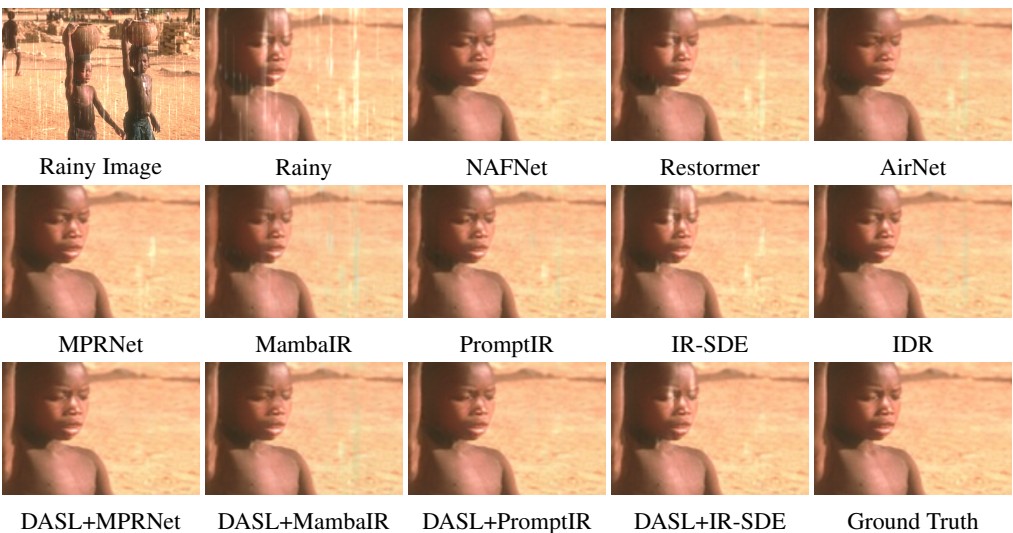

Figure 14: Visual comparison with state-of-the-art methods on Rain100L dataset.

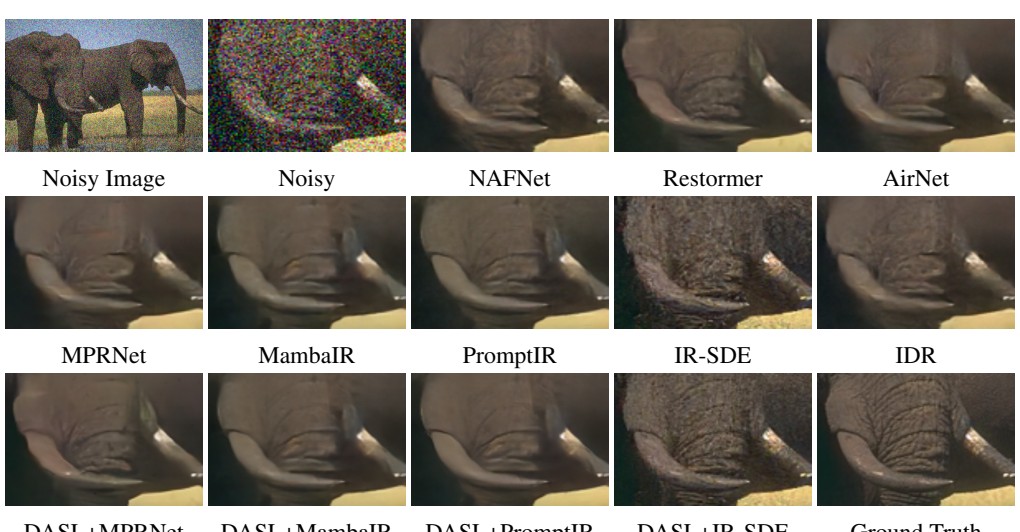

Figure 15: Visual comparison with state-of-the-art methods on BSD68 dataset.

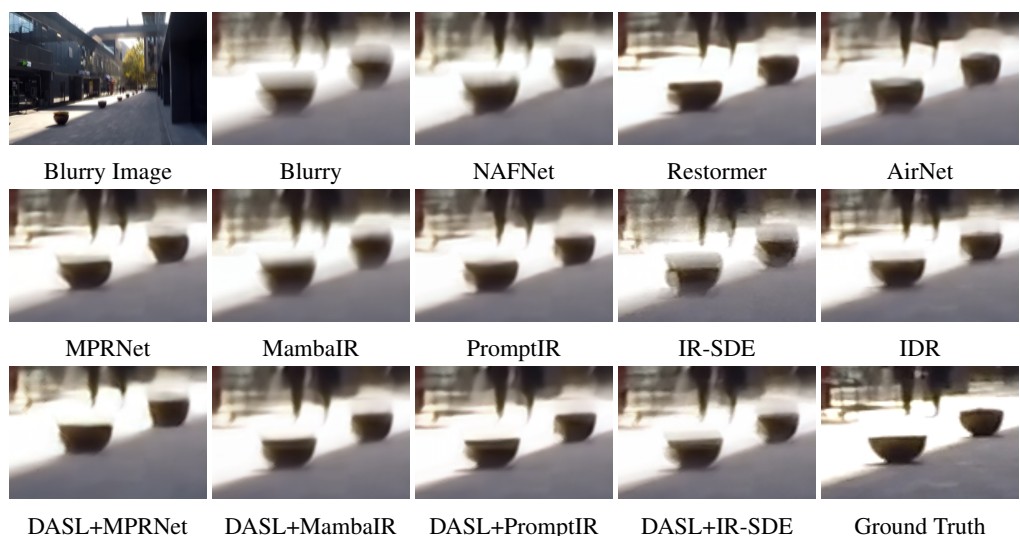

Figure 16: Visual comparison with state-of-the-art methods on GoPro dataset.

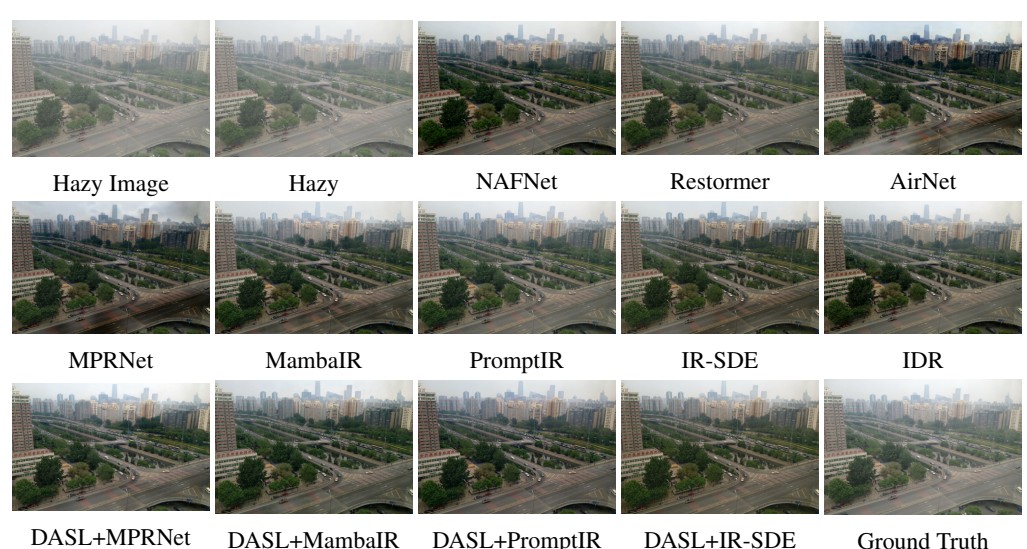

Figure 17: Visual comparison with state-of-the-art methods on SOTS dataset.

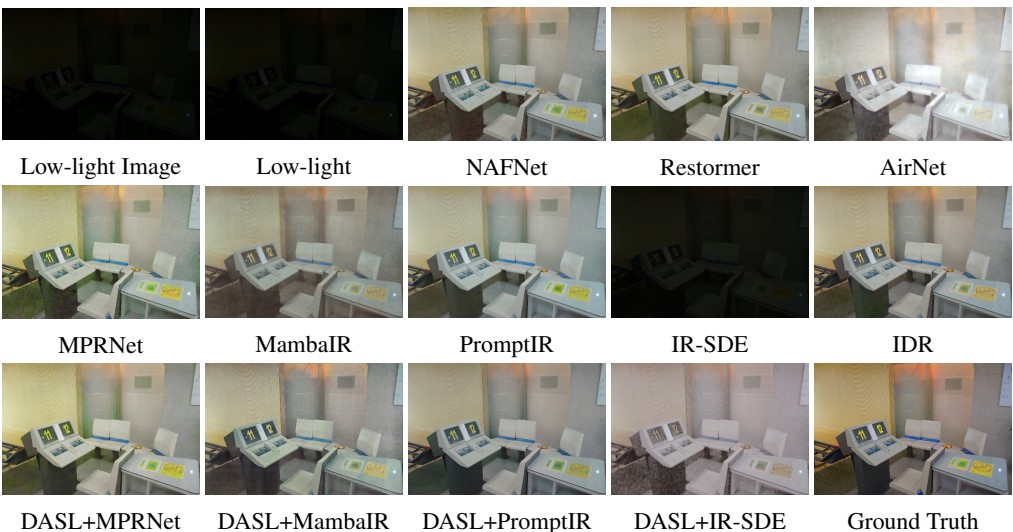

Figure 18: Visual comparison with state-of-the-art methods on LOL dataset.

