# OpenReview forum: "Decomposition Ascribed Synergistic Learning for Unified Image Restoration"
_ICLR.cc/2025/Conference — Submitted to ICLR 2025_

### Official Review · Reviewer_Eaz1 · 2024-11-03

**Soundness:** 2
**Presentation:** 3
**Contribution:** 2
**Rating:** 3
**Confidence:** 5

**Summary:**

The goal of this paper is to solve the image restoration task of multiple image degradation types. The paper explores the relationship between singular values and singular vectors between degraded images and clear images in different degradation types. Singular value-dominated and singular vector-dominated image restoration are ascribed. Singular vector operators and singular value operators are proposed and lightly integrated into the existing image restoration backbone. Experiments on multiple degradation types are conducted to verify the effectiveness of the method.

**Strengths:**

1. The paper is easy to understand and has a clear overall structure.
2. Exchange of the singular value and the singular vector analysis is performed on the image restoration tasks of different degradation types, showing the role of singular values ​​and singular vectors in image restoration.
3. The singular value operator and singular vector operator are proposed to make the existing methods lighter, with some improvement in computational complexity.

**Weaknesses:**

1. In Figure 2, the authors slightly show that the decomposed singular values ​​and singular vectors between different degradation types undertake different degradation information. Specifically, the exchange reconstruction error of Figure 2 (a) in the dehazing task is close to 50% for both singular value and singular vector, and the low-light and haze degradation in Figure 2 (b) are similar to other degradations such as rain and noise, while the blur and low-light in Figure 2 (c) are also very close. Therefore, I think that the authors don't explain the relationship between singular values ​​and singular vectors and degradation information well. The assumptions proposed are also untenable.
The authors could consider conducting an in-depth analysis in high-dimensional space or frequency space to analyze the impact of singular values ​​and singular vectors in terms of signal-to-noise ratio, frequency changes, etc. Even ablation analysis without using singular values ​​or singular vectors can be used instead of just statistical analysis of reconstruction errors.

2. The experiments only include results on a single degradation task dataset, not on a mixed dataset. This does not meet the requirement of using a single model for a unified image restoration task. The authors could refer to the similar experimental setup in the work of Li et al. [1] to conduct experiments on mixed degradation datasets to demonstrate the effectiveness and robustness of the proposed DASL.

3. The stability improvements brought by the loss curves of the training trajectories in Figure 6 and  Figure 7 are also not clearly demonstrated for tasks such as denoise and derain, etc. The authors should include the corresponding intermediate feature map changes or any other visualization statistics that can assist in demonstrating the effectiveness of singular values ​​and singular vectors.

[1] Li, Boyun, Xiao Liu, Peng Hu, Zhongqin Wu, Jiancheng Lv, and Xi Peng. "All-in-one image restoration for unknown corruption." In Proceedings of the IEEE/CVF conference on computer vision and pattern recognition, pp. 17452-17462. 2022.

**Questions:**

Same as described in the weaknesses.

---

> ### Author Response · Authors · 2024-11-21
> **Author response to Reviewer Eaz1**
>
> We thank the reviewer for your thoughtful comments and time.
>
> **Clarification of the illustration problem in Fig. 2.**
>
> Thanks for your kind reminder. There may exist some unclear illustration in Fig. 2 due to the numerical instability in pixel space. In Fig. 2(a), as we do not clip the range of the recomposed image between 0-255 for calculating the exchange reconstruction error, which may yield unexpected statistical results with numerical noise. Especially for the dehazing task, which is more prone for the recomposed image to yield the numerical overflow, owing to the more white (numerical higher) pixel value. In Fig. 2(b), we would like to note that the difference is more valuable for top singular values, which stand for the most signal energy. In Fig. 2(c), as the blur images are adopted from the GoPro dataset, and basically 2x larger than other degradation images, which may yield unfair comparison in calculating singular vectors difference. In the updated version of Fig. 2, we apply the range clipping to the recomposed image in Fig. 2(a), resizing different degradation images within the comparable scale in Fig. 2(b) and (c), and include more validation images (~5k) for illustration.
>
> **More in-depth analysis to illustrate the relationship between singular values ​​and singular vectors and degradation information.**
>
> Thanks for your suggestion. We would like to further illustrate the SVD-ascribed degradation analysis from the high-dimensional latent space. To be specific, we train a linear projector to transform the degraded latents to clean latents and compute the transmission ratio, where the clean latents are obtained with clean input and extracted from the same layer as degraded latents. To see how well the singular vectors and singular values represent the degradation information, two variants are conducted with firstly swapping clean singular vectors or singular values to degraded latents, and then perform the linear transformation. The results are shown in Tab. 7 in sec. 4.4, where the singular vector dominated degradations show significant transmission ratio improvement ($\sim$30%) when swapped with clean singular vectors, and exhibit minor transmission ratio improvement (~5%) when swapped with clean singular values. The same thing also happened in singular value dominated degradations with ~20%-50% transmission ratio improvement when swapped with clean singular values and ~3% transmission ratio improvement when swapped with clean singular vectors. Therefore, we have reason to suggest that the singular vectors and singular values could well distinguish the different types of degradation information, and the latent-space SVD-ascribed degradation analysis is consistency with pixel-space SVD-ascribed degradation analysis in ascribing degradation types.
>
> **Clarification on the experimental setting.**
>
> We would like to note that we do conduct experiments on mixed datasets and use a single model for unified image restoration. Beyond Li et al. which conduct experiments on three mixed degradation datasets, we conduct experiments on five mixed degradation datasets to validate the effectiveness of our method (Tab. 2), and reach improved performance compared to baseline method with accelerated inference time (Tab. 5). Additionally, we also include the experimental results on images with multiple degradation combination in Tab. 11 in Appendix D, and show noticeable performance gain.
>
> **The unclear loss curves in Fig. 6 and 7, and the effectiveness of SVEO and SVAO.**
>
> Thanks for your suggestion. We have reformed the loss curves in Fig. 6 and 7 with more sampled training steps to validate the training stability brought by SVEO and SVAO. The detailed ablation of SVEO and SVAO are further provided in Tab. 6, where the SVEO is more capable of boosting performance on singular vector dominated degradations and SVAO is more capable of boosting performance on singular value dominated degradations.
>
> Thanks for your efforts and hope our response and updates in the manuscript were able to address your concerns well. Please do let us know if you have any other questions.

---

> ### Author Response · Authors · 2024-12-02
> **Gentle Reminder to Reviewer Eaz1**
>
> We thank the reviewer for your effort and valuable comments. Would you mind checking the response to confirm if you have any further questions, and looking forward to your feedbacks.

---

### Official Review · Reviewer_9r6y · 2024-11-04

**Soundness:** 2
**Presentation:** 1
**Contribution:** 2
**Rating:** 5
**Confidence:** 4

**Summary:**

This paper mainly proposes Decomposition Ascribed Synergistic Learning (DASL) to restore multiple image degradations by SVD and FFT. Specifically, diverse degradations are ascribed into two groups, singular vector-dominated degradations and singular value-dominated degradations. The proposed DASL dedicates the decomposed optimization of them respectively, rendering a more unified perspective to inherently utilize the potential partnership among diverse restoration tasks for ascribed synergistic learning. Two effective operators SVEO and SVAO have been developed to favor the decomposed optimization. Experimental results demonstrate that DASL improves restoration quality while reducing model parameters and accelerating inference speed.

**Strengths:**

[S1] Lightweight networks have great application prospects at present.
﻿
[S2] DASL reduces the computations of the baseline model and improves the effect.

**Weaknesses:**

[W1] The paper writing is very poor, containing many typos. Many sentences, paragraphs, and sections are difficult to understand.

[W2] The proposed method applies the SVD in the FFT space and validates its effectiveness to some extent. However, it does not clearly explain the insight and why it works.

[W3] More visualization results of DASL+baseline should be presented rather than only MPRNet. In addition, in Fig. 14-16, the superiority of the proposed method is hard to distinguish.
﻿
[W4] The article presumes that the SVD and IDFT operate similarly in terms of signal formation, but does not show the Ablation Study of the restore results.

**Questions:**

[Q1] As per my understanding, the orthogonal regularization loss and decomposition loss in formula (5) represent the reconstruction loss of singular vectors and singular values respectively, and different dominant factors are employed for various types of degradation. During the training process, the balanced weights are set as a fixed ratio. Will this setting negatively harm the mixed results?
﻿
[Q2] In Figure 2, which baseline method did you use? It seems that the exchange reconstruction error for dehazing is very close. Thus, this figure can't convince me of the effectiveness of the proposed idea in dehazing task.
﻿
[Q3] This work aims to use a single model to handle multiple kinds of degraded images with one degradation type. How does it perform on images that contain multi-types of degradation? What is the difference in performance between your model and a model that trains only with a single type of degradation?

---

> ### Author Response · Authors · 2024-11-21
> **Author response to Reviewer 9r6y (1)**
>
> We thank the reviewer for your thoughtful comments and valuable feedback.
>
> **Clarification on applying SVD in FFT space, the insight and why works.**
>
> We would like to note that we do not apply SVD in FFT space. Instead, we perform the FFT to replace the SVD operation and approximate the singular values with fourier coefficients.
>
> The insight lies in the computation overhead, since the SVD computation is heavily time consuming, especially in high-dimensional latent space, directly acquiring the singular values with SVD computation in model optimization is unaffordable. Therefore, we resort to the FFT to approximate the singular values with fourier coefficients for optimization, which give us 516.8x acceleration as presented in Tab. 1.
>
> The reason why singular values can be approximated with fourier coefficients is attributed to their similar signal formation principle and detailed in sec3.3. Briefly speaking, both SVD and IDFT signal formation can be regarded as a weighted sum on a set of base components, as illustrated in Fig. 4, where singular values and fourier coefficients undertake the same role for linear combination of these base components, and can thus be approximated by each other.
>
> **More visual comparisons of DASL+baseline, and the visual superiority in Fig.14-16.**
>
> Thanks for pointing that out. We include more visual comparisons of DASL+baseline in Fig. 14-18, including MambaIR, PromptIR, and IR-SDE. We also update our results in Tab. 2-5 with newly added baselines and DASL integration.
>
> We note that the visual superiority in Fig.14-16 is basically noticeable and we would like to point this. In Fig.14, DASL provides more textured results after denoising compared to the smoothed version of baseline. In Fig.15, DASL significantly suppresses the jitter artifacts after deblurring compared to the baseline. In Fig.16, the visual improvement of DASL is globally with reduced shadow artifacts after dehazing compared to the baseline. Additionally, we refer the reviewer to see the updated Fig. 14-18 for more visual comparison.
>
> **The ablation of SVD and IDFT signal formation for restored results.**
>
> As presented before, the SVD formation is heavily time consuming, and requires 180.243s to perform the decomposition on tensor with size of 64x128x128, compared to 0.159s of FFT for the same size tensor, as illustrated in Tab. 1. Therefore, it is unaffordable to perform the SVD formation for ablation experiment, which is also why we need to resort to the FFT for model optimization.
>
> **Clarification on the meaning of orthogonal regularization loss and decomposition loss, and whether the fixed loss ratio harms the performance.**
>
> We would like to clarify that the reconstruction loss of singular vectors and singular values is formulated by decomposition loss $L_{dec}$ only, which is not related to the orthogonal regularization loss $L_{orth}$. The orthogonal regularization loss $L_{orth}$ is responsible for regularizing the weight parameters of SVEO to be orthogonal. Therefore, their loss ratios are not conflicted.
>
> We think what the reviewer concerns is whether the dynamic loss ratio should be adopted in $L_{dec}$ (Eq. 4) for balance between the singular vector regression and singular value regression, which is good question. Actually, different types of degradation natively have different loss values in singular vector regression and singular value regression, which is determined by their dominated degradation types, and there is no necessity to further adopt the dynamic loss ratio. What we need is to scale the loss value in $L_{dec}$ to be numerically comparable and the dynamic can be natively accomplished by loss value.
>
> **Which baseline used in Fig. 2.**
>
> We note that there is no baseline method involved in Fig. 2. The experiment in Fig. 2 is directly conducted in pixel space between degraded images and clean images, to statistically validate that the singular vectors and singular values undertake different types of degradation information.

---

> ### Author Response · Authors · 2024-11-21
> **Author response to Reviewer 9r6y (2)**
>
> **Clarification on the close exchange reconstruction error on dehazing task in Fig. 2(a).**
>
> We note that as we do not clip the range of the recomposed image between 0-255 in Fig. 2(a) for calculating the exchange reconstruction error, which may yield unexpected statistical results with numerical noise. Especially for the dehazing task, which is more prone for the recomposed image to yield the numerical overflow, owing to the more white (numerical higher) pixel value. In the updated version of Fig. 2, we apply the range clipping to the recomposed image and include more validation images (~5k) for illustration.
>
> **More convincing evidence for SVD-ascribed degradation analysis.**
>
> Beyond the pixel space, we would like to further illustrate the SVD-ascribed degradation analysis from the latent space. To be specific, we train a linear projector to transform the degraded latents to clean latents and compute the transmission ratio, where the clean latents are obtained with clean input and extracted from the same layer as degraded latents. To see how well the singular vectors and singular values represent the degradation information, two variants are conducted with firstly swapping clean singular vectors or singular values to degraded latents, and then perform the linear transformation. The results are shown in Tab. 7 in sec. 4.4, the singular value dominated degradations show significant transmission ratio improvement (26.7% for dehazing and 55.8% for low-light) when swapped with clean singular values, and exhibit minor transmission ratio improvement (3% for dehazing and 0.7% for low-light) when swapped with clean singular vectors, and the same thing also happened in singular vector dominated degradations with ~30% transmission ratio improvement when swapped with clean singular vectors and ~5% transmission ratio improvement when swapped with clean singular values. Therefore, we have reason to suggest that the singular vectors and singular values could well distinguish the different types of degradation information, and the pixel-space SVD-ascribed degradation analysis is consistency with latent-space SVD-ascribed degradation analysis in ascribing degradation types.
>
> **Performance on images contains multiple types of degradation.**
>
> We further include the experimental results on real-world under-display camera (UDC) image restoration in Tab. 11 in Appendix D for evaluating the multiple degradation combination. Typically, images captured under UDC system suffer from blurring due to the point spread function (PSF), and low light due to the lower display transmission rate. And DASL is also available in boosting the performance (~2dB) on such hybrid image restoration settings.
>
> **Performance comparison with models trained on single type of degradation.**
>
> We include the experiments with PromptIR below, including models trained on single type of degradation individually, model trained on mixed degradation, and DASL integrated model.
> |Methods|Rain100L|BSD68|GoPro|SOTS|LOL|Average|Params|
> |:-|:-:|:-:|:-:|:-:|:-:|:-:|-:|
> |PromptIR (single)|37.03 / 0.979|31.71 / 0.898|30.75 / 0.921|31.31 / 0.929|21.64 / 0.801|30.49/ 0.906| 35.59M|
> |PromptIR (mix)|34.24 / 0.957|31.30 / 0.885|26.43 / 0.802|25.18 / 0.934|21.69 / 0.805|27.76 / 0.876| 35.59M|
> |DASL+PromptIR (mix)|36.67 / 0.975|31.66 / 0.896|27.36 / 0.839|25.55 / 0.944|21.73 / 0.834|28.59 / 0.897|32.31M |
>
> The metrics are reported as PSNR/SSIM. Basically, the mix-trained model is hard to catch up with single-trained models, due to the limited model capability induced by limited model size and limited training data, and there is even no scaling laws in the visual field like that in language field for scaling up. Basically, DASL is capable of boosting the performance of the baseline method on such mix-trained case, and even reducing the model complexity with accelerated inference time.
>
> Thanks for your efforts and hope our response and updates in the manuscript were able to address your concerns well. Please do let us know if you have any other questions.

---

> ### Comment · Reviewer_9r6y · 2024-11-24
> **Response to authours**
>
> Thanks for the author's detailed response, which partially addresses my concerns.
>
> This work's main limitation lies in its motivation, mainly illustrated in Fig. 1. For example, in the tasks of denoising and deblurring, the visual examples cannot support the conclusion that the singular vectors dominate the degradation. Furthermore, it also lacks some theoretical foundation.
>
> For the SVAO proposed in Sec. 3.3, I capture the author's insights. Considering SVD and IDFT are both formulated as a linear combination with a set of bases, thus the author adopts an IDFT to approximate SVD. However, it is necessary to explain that optimizing the amplitude maps is equal (or at least has a very similar effect) to optimizing the singular values.
>
> Based on these two pivotal concerns, I still keep my negative rating.

---

> > ### Author Response · Authors · 2024-12-02
> > **Gentle Reminder to Reviewer 9r6y**
> >
> > We thank the reviewer for your effort and valuable comments. Would you like further checking the response to confirm if you have any further questions, and looking forward to your feedbacks.

---

> > > ### Comment · Reviewer_9r6y · 2024-12-03
> > > **Response to authors.**
> > >
> > > Thanks for the author's response. However, such a subjective explanation for the second concern still cannot convince me. I thus keep my score.

---

> ### Author Response · Authors · 2024-11-24
> **Response to Reviewer 9r6y**
>
> Thanks for your timely feedback and reviewing efforts. We would like to further illustrate on your concerns.
>
> **Concern about the visual examples in Fig. 1 of the denoising and deblurring tasks, and theoretical foundation.**
>
> In Fig. 1, we think what the reviewer concerns is the damaged pixels exist in singular vector swapped results in the denoising and deblurring task. We would like to clarify that the damaged pixels of the recomposed image is due to the numerical overflow, which can be simply avoided by clipping the range of the recomposed image between 0-255. We do not do this because we would like to show the native recomposition results in the spirit of first principle to exhibit how well the singular vectors and singular values undertake different types of degradation information. Actually, such numerical overflow is more likely to appear in more white (numerical higher) pixel values of the input image, and is not related to the degradation information. Regardless of such numerical factor, we could see that the singular vector swapped results of the denoising and deblurring show more clean scene content and suppressed blur effects. More visual results are provided in Fig. 11 and 12. Actually, we also provide the compound degradation analysis including blurry+noisy case in Fig. 13, which may be more evident as we clip the range of the recomposed image between 0-255 to exclude the numerical factor for visualization.
>
> We provide more in-depth analysis of SVD-ascribed degradation analysis in Appendix G, including the theoretical foundation. Briefly speaking, from the signal formation perspective, any signal can be regarded as a weighted sum on a set of basis, i.e., $X=U\Sigma V^T=\sum_{i=1}^{k}\sigma_{i}u_{i}v_{i}^T$. In this light, the singular vectors $\cup_{i=1}^{k}\lbrace u_{i}v_{i}^T\rbrace$ (2D matrices) represent the base components of the signal for content composition, and the singular values $\cup_{i=1}^{k} \lbrace\sigma_{i}\rbrace$ (scalers) represent the combined coefficients of the base components for statistical modulation. Therefore, their respective dominated degradation types are fundamentally determined by such signal formation principle, i.e., content corruption (rain, noise, blur) and statistic distortion (hazy, low-light).
>
> **Concern that optimizing the amplitude maps is approximate to optimizing the singular values.**
>
> We are grateful to know that the reviewer has captured our insights on the similar signal formation principle of SVD and IDFT. We would like to be more specific, as declared in sec. 3.3. In the SVD signal formation, i.e., $X=U\Sigma V^T=\sum_{i=1}^{k}\sigma_{i}u_{i}v_{i}^T$, the singular vectors $\cup_{i=1}^{k}\lbrace u_{i}v_{i}^T\rbrace$ represent the base components of the signal, and the singular values $\cup_{i=1}^{k} \lbrace\sigma_{i}\rbrace$ represent the combined coefficients of the base components. In the IDFT signal formation, i.e., $X = \frac{1}{hw}
> \sum_{u=0}^{h-1}\sum_{v=0}^{w-1} G(u,v)e^{j2\pi(\frac{um}{h}+\frac{vn}{w})}$, the phase term $\cup_{u=0,v=0}^{h-1,w-1}\lbrace e^{j2\pi(\frac{um}{h}+\frac{vn}{w})} \rbrace$ (2D matrices) represent the base components of the signal, and the amplitude term $\cup_{u=0,v=0}^{h-1,w-1}\lbrace G(u,v) \rbrace$ (scalers) represent the combined coefficients of the base components. Therefore, we have reason to suggest that the singular values and amplitude maps undertake the same role for linear combination of the base components, and can be approximated with each other for optimization. Besides, to be more specific, the low-frequency amplitude correspond to top singular values, and the high-frequency amplitude correspond to tail singular values, owing to their similar signal energy distribution, as shown in Fig. 5.

---

> ### Author Response · Authors · 2024-12-03
> **Response to Reviewer 9r6y**
>
> Thanks for your feedback. We would like to note that the approximation between singular values and fourier coefficients (i.e., amplitude map) has been theoretically verified in last response and sec. 3.3. And we could see that both singular values and fourier coefficients undertake the same role in signal formation, i.e., linear coefficients for combining base components, e.g., singular vectors or fourier phase term (2D wave). The main difference is that the singular vectors are content-related base components, and the fourier phase term are content-agnostic base components, however, both conform to the same component construction principle, i.e., from outline to details, as shown in Fig. 5, indicating that they share the similar signal energy distribution with principal skew rather than uniform distribution. Therefore, we could reach the approximation relationship between singular values and fourier coefficients.
>
> We think what the reviewer concern is the ablation results of SVD optimization and IDFT optimization. However, as we provided in first reply, explicitly performing  the SVD in high-dimensional tensor for optimization is extremely time-consuming, and requiring 180.243s on tensor with size of 64x128x128, compared to 0.159s of FFT for the same size tensor. Therefore, it is unaffordable to perform such ablation for SVD optimization, and we hope the reviewer could take this into consideration. Actually, any explicit SVD in high-dimensional latent space could be unaffordable, which is exactly the intention of our DASL proposed for, i.e., utilizing the property of SVD-ascribed degradation analysis, and avoiding perform the explicit SVD in optimization. The ablation of SVAO in Tab. 6 suggesting that it is well responsible for singular value dominated degradations, which could partially illustrate that the fourier coefficients optimization well approximates the singular values optimization.

---

### Official Review · Reviewer_R1kv · 2024-11-04

**Soundness:** 3
**Presentation:** 2
**Contribution:** 3
**Rating:** 5
**Confidence:** 4

**Summary:**

The manuscript explored the restoration of multiple image degradations through a unified model. By employing singular value decomposition, the authors classified restoration tasks into singular vector dominated and singular value dominated categories. They introduced the Decomposition Ascribed Synergistic Learning (DASL) framework, which optimized the decomposed components to exploit relationships among various restoration tasks. The framework included two operators, Singular VEctor Operator (SVEO) and Singular VAlue Operator (SVAO), along with a supporting decomposition loss.

**Strengths:**

This paper addresses a significant gap in the field by proposing a unified approach to handling multiple image degradations, moving beyond the conventional practice of treating each degradation in isolation.

The use of singular value decomposition to categorize restoration tasks into singular vector dominated and singular value dominated groups provides a fresh perspective on the relationships between different types of degradations.

**Weaknesses:**

The manuscript raises several important questions regarding its methodology and results that require clarification.

The current description lacks clarity.

**Questions:**

Are there specific references supporting the findings presented in Figures 1 and 2? Additionally, clarification is needed on the type of Singular Value Decomposition (SVD) employed in the analysis and whether any post-processing techniques were applied to produce the displayed images. It is noted that the deblurring case exhibits a relatively large area of damaged pixels, while the hazy and low-light image enhancement cases only show minor pixel degradation. What factors contribute to this discrepancy?

The authors conducted tests on underwater enhancement and sandstorm enhancement but did not include these results in Figure 1 & 2. Are the observations in Figure 1 commonly encountered across different datasets, including underwater and sandstorm? Moreover, Figure 2 is based on 100 images; Additional images should be tested to validate the findings. It is unclear whether only synthetic images were used to generate Figures 1 and 2, such as in the rainy case.

The assumptions regarding singular vectors capturing content information and spatial details, and singular values representing global statistical properties, are questionable. The authors should provide more supporting references for these claims. Furthermore, given that degradations are categorized into two groups, does the proposed method require prior knowledge of which group the degradation falls into before reconstruction?

A clear diagram illustrating the entire blueprint of the proposed methods should be included to demonstrate how the SVEO and SVAO are integrated into the algorithm. The current description lacks clarity, particularly the statement about substituting convolution layers with SVEO.

The experiments primarily report PSNR results; however, additional metrics should also be considered to provide a more comprehensive evaluation of the proposed method's performance.

The rationale for using the L1 norm in L_dec should be clarified.

Figure 5 does not effectively illustrate progressive component reconstruction as intended.

Typos in the abstract: 'tasks into two groups, , '

---

> ### Author Response · Authors · 2024-11-21
> **Author response to Reviewer R1kv (1)**
>
> We thank the reviewer for your thoughtful comments and valuable feedback.
>
> **Are there any references for findings in Fig.1 and 2, and clarification on the type of SVD employed.**
>
> We would like to note that we are the first to propose the findings that the singular vectors and singular values naturally undertake different types of degradation information, which has not appeared in the literature and is our native contribution. We provide deeper analysis in Appendix G, to support the reasonability of SVD-ascribed degradation analysis, with detailed theoretical analysis from signal formation perspective and more generalization experiments.
>
> We perform the basic full SVD in our analysis experiments without truncation or randomization. The only operation in producing images in Fig. 1 is to swap the singular vectors and singular values between clean images and degraded images, and there is no post-processing applied.
>
> **Why damaged pixels in Fig. 1 and the size of damaged region.**
>
> The damaged pixels are caused by numerical overflow, and can be simply avoided by clipping the range of the recomposed image between 0-255. We do not do this because we would like to show the native recomposition results in the spirit of first principle to exhibit how well the singular vectors and singular values undertake different types of degradation information. The size of damaged region is directly related to the size of white (numerical higher) pixel values of the input image, which are more prone to yield the numerical overflow.
>
> **Whether the observations in Fig.1 are commonly encountered for datasets in appendix.**
>
> Yes, we note that there is no meaning that the degradations listed in the appendix are less conformed to the SVD-ascribed degradation analysis than degradations listed in the main text, and we believe the experiments in Fig. 9 and 10 have illustrated this well.
>
> Actually, we provide the verification in Appendix G to theoretically illustrate whether the SVD-ascribed degradation analysis is generalizable. From the SVD signal formation, any signal can be regarded as a weighted sum on a set of basis, i.e., $X=U\Sigma V^T=\sum_{i=1}^{k}\sigma_{i}u_{i}v_{i}^T$. The generalization is thus verified via such closed form of signal formation with singular vector dominated degradations $\cup_{i=1}^{k}\lbrace u_{i}v_{i}^T\rbrace$ and singular value dominated degradations $\cup_{i=1}^{k} \lbrace\sigma_{i}\rbrace$.
>
> **More images for validation in Fig. 2, and whether the SVD-ascribed degradation analysis is usable for real case.**
>
> Thanks for pointing that out. We have updated more images (~5k) in Fig. 2 to perform the statistic validation, and the results are consistent with reduced variance. The updated Fig. 2 incorporates 50% of real rainy images from [SPANet; CVPR2019] dataset, and the remaining degradations were originally containing real cases. We include the degradation analysis for real images (e.g., rainy) in Fig. 11 in appendix, and the observation are consistency.
>
> We would like to note that the SVD-ascribed degradation analysis is robust to the real/synthetic domain gap, as the singular vector/value gap ascribed by SVD-analysis is more governing and fundamental, which directly impacts the decomposed signal components rather than the whole signal.
>
> **References for why singular vectors represent content information and singular values represent statistics.**
>
> We would like to note that we are the first to propose this finding which is our native contribution, and there are no references to claim this to our knowledge. Actually, we draw this conclusion experimentally and theoretically, and the detailed analysis is provided in Appendix G. To be specific, from the of signal formation perspective, any signal can be regarded as a weighted sum on a set of basis, i.e., $X=U\Sigma V^T=\sum_{i=1}^{k}\sigma_{i}u_{i}v_{i}^T$. In this light, the singular vectors $\cup_{i=1}^{k}\lbrace u_{i}v_{i}^T\rbrace$ (vectors) represent the base components of the signal for content composition, and the singular values $\cup_{i=1}^{k} \lbrace\sigma_{i}\rbrace$ (scalers) represent the combined coefficients of the base components for statistical modulation. Therefore, their respective dominated degradation types are fundamentally determined by such signal formation principle, i.e., content corruption (rain, noise, blur, low-resolution, JPEG compression, color-shifting) and statistical distortion (hazy, low-light, underwater enhancement, sandstorm enhancement).

---

> ### Author Response · Authors · 2024-11-21
> **Author response to Reviewer R1kv (2)**
>
> **Whether the proposed method needs to know the ascribed degradation type before reconstruction.**
>
> No, we would like to note that there is no need for the model to acquire the ascribed degradation type before reconstruction. In DASL, the optimization for singular vectors and singular values are both conducted through SVEO and SVAO, and we do not turn off one of them when encountering another type of degradation, which is also why DASL is able to handle hybrid degradation existing in one image, as presented in Tab. 11 in Appendix D.
>
> **Illustrating how SVEO and SVAO are integrated into the baseline method.**
>
> We note that the architecture of baseline methods can be different, and it is hard to draw a unified macro blueprint to illustrate how SVEO and SVAO are integrated. Instead, we would like to illustrate this from a micro perspective, which we directly reform the basic block of the backbone network with our operator, as presented in Fig. 8 in Appendix B, regardless of diversified model architectures. To be more specific, the integration of SVEO is accomplished by substituting half of the activated convolution layers in basic block, and the SVAO is only enabled at substituted basic blocks in bottleneck layers.
>
> **The experiments primarily report PSNR, and more metrics for evaluation.**
>
> Thanks for your suggestions. The main evaluation in our experiments includes performance metrics PSNR, SSIM and complexity metrics model size, flops and inference time, and we further include the LPIPS in Appendix D for evaluation.
>
> **The rationale of using $L_1$ norm in $L_{dec}$.**
>
> We note that the $L_{dec}$ in Eq. 4 can be simply regarded as a regression loss that we typically use in image reconstruction tasks, where the $L_1$ norm and $L_2$ norm are broadly used to close the gap between the model output and groundtruth. Therefore, the rationale of $L_1$ norm in $L_{dec}$ is equivalent to $L_1$ norm in typical image regression loss. The only difference is that previous regression loss closes the gap on the whole signal, while the $L_{dec}$ closes the gap on the decomposed signal, i.e., singular vectors and singular values.
>
> **Clarification on the progressive component reconstruction in Fig. 5.**
>
> Thanks for pointing this out. We would like to note that the progressive component reconstruction cannot be linearly in Fig. 5, since the signal energy is primarily distributed on the principal components of the signal, i.e., the low-frequency in IDFT signal formation and the top singular components in SVD signal formation. Fig.5 intends to illustrate that the IDFT signal formation and SVD signal formation enjoy the almost synchronous signal formation character from outline to details, and thus we could approximate singular values with fourier coefficients for accelerated FFT optimization.
>
> Thanks for your efforts and hope our response and updated version were able to address your concerns well. Please do let us know if you have any other questions.

---

> > ### Author Response · Authors · 2024-12-02
> > **Gentle Reminder to Reviewer R1kv**
> >
> > We thank the reviewer for your effort and valuable comments. Would you mind checking the response to confirm where you have any further questions, and looking forward to your feedbacks.

---

### Official Review · Reviewer_8UnW · 2024-11-04

**Soundness:** 3
**Presentation:** 3
**Contribution:** 3
**Rating:** 6
**Confidence:** 4

**Summary:**

The paper introduces Decomposition Ascribed Synergistic Learning (DASL), a method for unified image restoration that optimizes singular vectors and values for handling multiple degradations within a single model, and integrates seamlessly into existing restoration architectures.

**Strengths:**

1. The paper uncovers an observation that the decomposed singular vectors and values naturally undertake the different types of degradation information, ascribing various restoration tasks into two groups, i.e., singular vector dominated and singular value dominated.
2. Two operators are developed to favor the decomposed optimization of degraded singular vectors and values for various restoration tasks.

**Weaknesses:**

1. There is a limited evaluation with recent AIR methods like PromptIR, GenLV, and the diffusion-based AIR techniques. Such comparisons ensure the research is aligned with the latest advancements in the field.
2. Do the observations regarding the decomposition of singular vectors and singular values hold true in terms of other degradations (low-resolution, jepg compression, etc.) or compound degradations (noise+blur, rain+haze, etc.)? Whether the proposed method is available in multiple degradation combinations?
3. How well do the decomposed singular vectors and singular values represent degradation information, and are they discriminative enough to distinguish between different degradations?
4. Some typos like ", ,".

**Questions:**

Please see the above Weaknesses.

---

> ### Author Response · Authors · 2024-11-21
> **Author response to Reviewer 8UnW**
>
> We thank the reviewer for your thoughtful comments and time.
>
> **Comparison with latest advancements in the field.**
>
> Thanks for your kind reminder. We further include the latest methods below, and thoroughly update the results in the manuscript (Tab. 2-5, Fig. 14-18.). As the GenLV is not open sourced yet, we replace it with DA-CLIP for comparison.
>
> |Methods|Rain100L|BSD68|GoPro|SOTS|LOL|Average|Params|
> |:-|:-:|:-:|:-:|:-:|:-:|:-:|-:|
> |PromptIR|34.24 / 0.957|31.30 / 0.885|26.43 / 0.802|25.18 / 0.934|21.69 / 0.805|27.76 / 0.876| 35.59M|
> |IR-SDE|35.18 / 0.969|30.26 / 0.895|25.63 / 0.777|24.73 / 0.925|11.83 / 0.473|25.53 / 0.808| 137.15M|
> |DA-CLIP|35.69 / 0.974|30.45 / 0.898|25.92 / 0.786|25.24 / 0.938|17.96 / 0.738|27.05 / 0.867|136.82M|
> |DASL+PromptIR|36.67 / 0.975|31.66 / 0.896|27.36 / 0.839|25.55 / 0.944|21.73 / 0.834|28.59 / 0.897|32.31M |
> |DASL+IR-SDE|35.46 / 0.972|30.43 / 0.901|25.91 / 0.789|25.08 / 0.941|15.26 / 0.614|26.42 / 0.843| 128.64M|
> |DASL+DA-CLIP|35.78 / 0.979|30.87 / 0.901|26.08 / 0.789|25.53 / 0.947|19.21 / 0.753|27.49 / 0.874|130.45M|
>
> The metrics are reported as PSNR/SSIM, where DASL brings consistent performance gain across diverse architectures.
>
> **Do the SVD-ascribed degradation analysis hold true on other degradations, or compound degradations?**
>
> Yes, we include more other types of degradation analysis in Fig.9 and 10 in Appendix G, including low-resolution, JPEG compression, color-shifting, underwater enhancement, and sandstorm enhancement, where the former three types are ascribed into singular vector dominated degradations and the latter two types are ascribed into singular value dominated degradations.
>
> The SVD-analysis on compound degradations are further provided in Fig. 13, including noise+rain, noise+blur, rain+blur, noise+low-light and rain+haze, and are consistent with what we ascribed in single degradation-analysis. To be specific, the former three types are ascribed as homogeneous compound degradations with both degradation information identified at one side of singular vectors or singular values, and the latter two types are ascribed as heterogeneous compound degradations with degradation information identified at both sides of singular vectors and singular values.
>
> **Whether the method is available in multiple degradation combinations?**
>
> Yes, we include the experimental results on real-world under-display camera (UDC) image restoration in Tab. 11 in Appendix D for evaluating multiple degradation combinations. Typically, images captured under UDC system suffer from blurring due to the point spread function (PSF), and low light due to the lower display transmission rate. And DASL is also available in boosting the performance on such hybrid image restoration settings (~2dB improvement).
>
> **How well do the singular vectors and singular values distinguish the degradation information?**
>
> We would like to illustrate this from the latent space beyond the pixel space. In sec.4.4, we train a linear projector to transform the degraded latents to clean latents and compute the transmission ratio, where the clean latents are obtained with clean input and extracted from the same layer as degraded latents. To see how well the singular vectors and singular values represent the degradation information, two variants are conducted with firstly swapping clean singular vectors or singular values to degraded latents, and then perform the linear transformation. The results are shown in Tab. 7 in sec. 4.4, where the singular vector dominated degradations show significant transmission ratio improvement ($\sim$30%) when swapped with clean singular vectors, and exhibit minor transmission ratio improvement ($\sim$5%) when swapped with clean singular values. The same thing also happened in singular value dominated degradations with $\sim$20%-50% transmission ratio improvement when swapped with clean singular values and ~3% transmission ratio improvement when swapped with clean singular vectors. Therefore, we have reason to suggest that the singular vectors and singular values could well distinguish the different types of degradation information
>
> Thanks for your effort and hope our response and updates in manuscript were able to address your concerns well.

---

> > ### Author Response · Authors · 2024-12-02
> > **Gentle Reminder to Reviewer 8UnW**
> >
> > We thank the reviewer for your effort and valuable comments. Would you mind checking the response to confirm where you have any further questions, and looking forward to your feedbacks.

---

### Author Response · Authors · 2024-11-21
**Official Comment by Authors**

We are sincerely grateful to the reviewers for dedicating their time and effort to review our work, and we appreciate the recognition of the fresh perspective of degradation relationship raised by this work, and the clear presentation structure. We have made numerous updates to the submission, most notably with the results of our DASL integration on new baselines and more validation of SVD-ascribed degradation analysis.

1. We include more latest methods in our experiments and update the results in Tab. 2-5, Fig. 14-18.

2. We include more validation images (~5k) in Fig. 2, and apply post-processing to the recomposed image to avoid numerical overflow for stable statistical illustration.

3. We include more sampled training steps in Fig. 6 and 7 to validate the training stability brought by SVEO and SVAO.

4. We include the SVD-ascribed degradation analysis on compound degradations in Fig. 13 for generalization verification.

5. We would like to raise the reviewer’s attention that the SVD-ascribed degradation analysis is not only applicable in pixel space, but also usable in high-dimensional latent space, as indicated in Tab. 7.

We also address all specific questions, comments, and concerns raised by reviewers in our detailed individual responses for each reviewer.

---

### Meta-Review · Area_Chair_sMA4 · 2024-12-19

**Metareview:**

This paper proposes a decomposition ascribed synergistic learning approach for unified image restoration. It is mainly based on the observation that the decomposed singular vectors and singular values are related to the different types of degradation. Experimental results show the effectiveness of the proposed method.

This paper received reviews with significant divergent ratings. The major concerns include the motivation, presentation, and evaluations.

The authors provide the rebuttal to solve the concerns of reviewers. The rebuttal solves some concerns well. For example, the authors provide more comparisons with the suggested methods by Reviewer 8UnW. However, the motivation and technique details are not explained well. For the provided comparisons (single v.s. mixed), the model on the mixed degradations does perform well compared to the one on the single degradation (about 2dB lower based on the results in the rebuttal).

Based on the recommendations of reviewers, the paper is not accepted.

**Additional Comments On Reviewer Discussion:**

In the discussions, reviewers still have concerns about the motivation and technique details. In addition, the performance on the mixed degradations is not good.

---

### Decision · Program_Chairs · 2025-01-22

Reject